# Features of acute COVID-19 associated with post-acute sequelae of SARS-CoV-2 phenotypes: results from the IMPACC study

Post-acute sequelae of SARS-CoV-2 (PASC) is a significant public health concern. We describe Patient Reported Outcomes (PROs) on 590 participants prospectively assessed from hospital admission for COVID-19 through one year after discharge. Modeling identified 4 PRO clusters based on reported deficits (minimal, physical, mental/cognitive, and multidomain), supporting heterogenous clinical presentations in PASC, with sub-phenotypes associated with female sex and distinctive comorbidities. During the acute phase of disease, a higher respiratory SARS-CoV-2 viral burden and lower Receptor Binding Domain and Spike antibody titers were associated with both the physical predominant and the multidomain deficit clusters. A lower frequency of circulating B lymphocytes by mass cytometry (CyTOF) was observed in the multidomain deficit cluster. Circulating fibroblast growth factor 21 (FGF21) was significantly elevated in the mental/cognitive predominant and the multidomain clusters. Future efforts to link PASC to acute anti-viral host responses may help to better target treatment and prevention of PASC.

Post-acute sequelae of SARS-CoV-2 (PASC), also known as long coronavirus disease 2019 (long COVID-19), represents a growing concern in public healthcare. While consensus for the clinical definition of PASC is evolving, the National Institute for Health and Care Excellence (NICE) defines PASC as signs and symptoms that develop during or following an infection consistent with COVID-19, continue for more than four weeks and are not explained by an alternative diagnosis[1,2]. Several prospective cohorts of PASC have been described, each differing in case definition, size, and composition of the study population, symptoms evaluated, as well as follow-up frequency and duration. The prevalence of PASC is especially high amongst those COVID-19 patients who needed hospitalization: according to some reports[3], up to half of hospitalized patients reported at least one physical, cognitive, or mental impairment, months after COVID-19 diagnosis. PASC is, thus, having a substantial impact on quality of life, healthcare costs, and economic productivity[4].

The immunobiology of PASC is currently under intensive investigation with some leading hypotheses[5] invoking the persistence of viral components driving immune stimulation, reactivation of viral infections such as EBV, dysbiosis of microbiome or virome, unrepaired tissue damage, and autoimmunity[6,7].

Here, we describe patient-reported recovery data prospectively assessed from the acute infection through one year after hospital discharge for COVID-19. We identify predisposing factors and immune profiles from the acute phase of the disease that are associated with impaired clinical and functional recovery during the year following hospital discharge.

## Results

### Demographics and descriptive statistics

In all, 1164 participants were enrolled between May 5th, 2020 and March 19th, 2021 and followed up to 28 days while hospitalized. Of the 702 participants who survived hospitalization and were alive and on study at 3 months post discharge, 590 (84%) completed at least one quarterly set of surveys post discharge (survey respondent cohort) (Fig. 1A) with 29% (170) completing all 4 quarterly surveys and most completing 2 or more surveys (494; 84%)[7]. The participants who responded appear to be demographically representative

✉ e-mail: nroupha@emory.edu

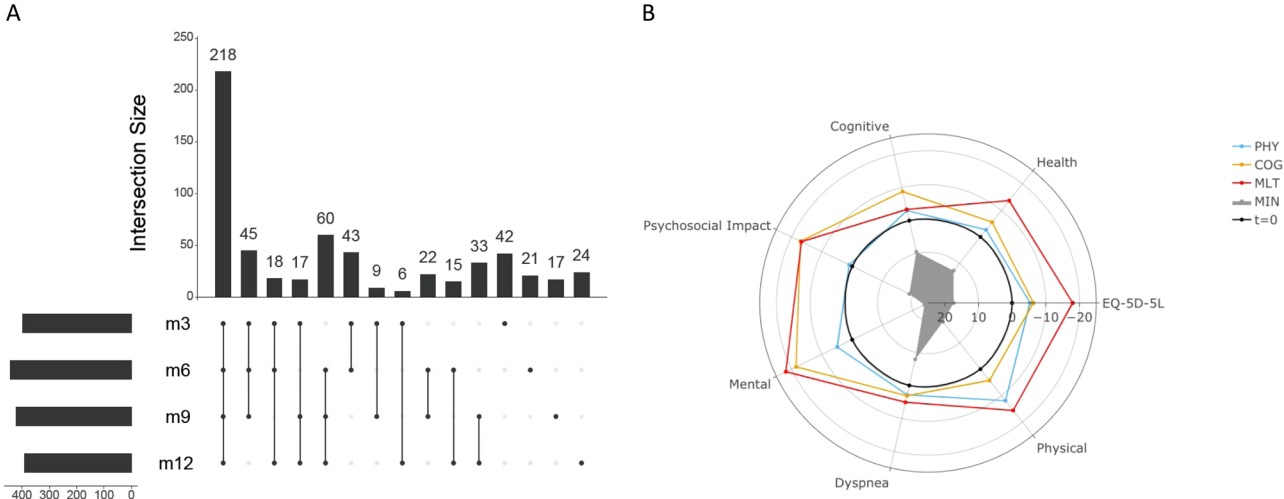

**Fig. 1 | Survey completion and clustering of participant-reported outcomes after hospital discharge (*N* = 590). A** Upset plot depicting the number of participants completing surveys at 3 (m3), 6 (m6), 9 (m9), and 12 months (m12) after hospital discharge. **B** Radar plot showing relative deficit for each of four different clusters across several participant-reported outcomes: EQ-5D-5L Health Recovery Score (Health), PROMIS Cognitive Function Score (Cognitive), PROMIS Psychosocial Illness Impact Positive Score (Psychosocial), PROMIS Global Mental Health Score (Mental), PROMIS Dyspnea Score (Dyspnea), and PROMIS Physical Function Score (Physical). The radial axis denotes a t-statistic comparing the within-cluster mean to the remaining sample, with $t = 0$ denoting the overall sample mean and negative values denoting a deficit. The 4 clusters are: solid gray, minimal deficit (MIN); blue line, physical predominant deficit (PHY); yellow line, mental/cognitive predominant deficit (COG); and red line, multidomain deficit (MLT). PROMIS Patient-Reported Outcomes Measurement Information System.

of the entire cohort but were less likely to have prolonged hospitalization or discharge limitations compared with non-responders (42% versus 57%)[8]. Demographics, clinical characteristics, baseline radiographic and laboratory findings, as well as main outcomes during the COVID-19 hospital stay are provided for the survey respondent cohort in Table 1. The median age was 57 years (IQR 19), and 360 (61%) were men. 131 (22%) were Black/African American, and 189 (32%) were Hispanic/Latinx. 94% of participants had at least one comorbidity, most commonly hypertension (327; 55%) and diabetes (200; 34%). The median body mass index (BMI) was 31.8 kg/m² (IQR 27.4–37.0). Two hundred and forty-five (52%) had an elevated baseline C-reactive protein (CRP) (≥10 mg/L) and 297 (50%) had an abnormal baseline D-dimer (>0.5 mg/L) upon hospital admission. Only 137 (25%) had no infiltrate on chest imaging upon hospital admission. One hundred and forty-one (24%) did not receive any oxygen therapy and 159 (27%) received ICU level care while inpatient. Four hundred (68%) received steroids and 377 (64%) received remdesivir. The median length of stay was 6 days (IQR 4–10) and most study participants had at least one complication (490; 83%) while inpatient. None of the participants had received a COVID-19 vaccine prior to admission; after discharge, 62% reported receiving the primary vaccine series and 36% a single booster dose. Three hundred and five (52%) out of 590 reported at least one symptom during the quarterly surveys, most commonly dyspnea (29%), followed by muscle aches/myalgia (21%), cough (20%), headache (19%), and fatigue/malaise (18%) (Table 1) (Figs. 1S and 2S). Thirty-two percent had symptoms affecting more than one organ system.

### Latent class and cluster analysis
Fitting latent class mixed models (LCMMs) derived from the longitudinal patterns to each of eight participant-reported outcomes (PROs), we selected quadratic models with three groupings for EQ-5D-5L, Health Recovery Score, and PROMIS Dyspnea Score, and a linear model with three groupings for PROMIS Cognitive Function Score. There were no distinct groupings for the PROMIS Physical Function Score, PROMIS Global Mental Health Score, and PROMIS Psychosocial

Illness Impact Positive Score. Median (IQR) values by visit for each PRO grouping are shown in Fig. 3SA–G.

We identified the Ward algorithm with six clusters as the optimal model. Fitting statistics across the five algorithms are shown in Fig. 3SH. Comparison of t-statistics across the PROs identified no associated deficit with certain clusters, which we then collapsed into a single cluster which we labeled minimal deficit (cluster MIN, 358 participants, 60.7%, Table 1A–S). Based on associations with specific PROs, clinical phenotypes were defined by labeling the remaining three clusters as physical predominant (cluster PHY, 92 participants, 15.6%), mental/cognitive predominant (cluster COG, 82 participants, 13.9%), and multi/pan domain deficit (cluster MLT, 58 participants, 9.8%). Table 1B–S and Fig. 1B, respectively, show a table and radar plot with the t-statistics for each PRO across these four clusters, which were named based on the predominant deficit for a specific PRO.

Selected demographic characteristics, key comorbidities, and laboratory findings were significantly associated with the four PRO clusters by bivariate analysis (Table 1). Acute phase disease severity, whether defined by respiratory score at admission, SOFA score at admission, ICU utilization, mechanical ventilation, or inpatient overall clinical trajectory, was not associated with PRO cluster assignment (Table 1). Any use of remdesivir and steroids in the inpatient period was not associated with a decrease in PASC prevalence (Table 1). Adjusted multinomial logistic regression analyses comparing to participants with the minimal deficit cluster indicated that participants in the PHY cluster were more likely to have reported comorbidities of chronic pulmonary disease (OR 2.46 95% CI 1.41–4.29) or chronic neurologic disorder (OR 2.13 95% CI 1.20–3.78) and less likely to be males (OR 0.55; 95% CI 0.35–0.87) and non-white race (OR 0.66; 95% CI 0.47–0.93). Relative to participants in the MIN cluster, participants with mental/cognitive predominant deficit were less likely to be 65 years or older (OR 0.41; 95% CI 0.18–0.94), less likely to be males (OR 0.54; 95% CI 0.36–0.82), more likely to have chronic cardiac disease (OR 1.72; 95% CI 1.02–2.88) and had longer acute hospitalization (OR per week 1.28; 95% CI 1.12–1.46). Participants with MLT deficit were more likely to have chronic pulmonary disease (OR 1.78; 95% CI 1.01–3.13) or chronic neurologic disorder (OR 4.37; 95% CI 2.14–8.94),

**Table 1 | Baseline characteristics of the survey respondent cohort**

| | | Overall (n = 590) | Cluster MIN (n = 358, 60.7%) | Cluster PHY (n = 92, 15.6%) | Cluster COG (n = 82, 13.9%) | Cluster MLT (n = 58, 9.8%) | Overall p value |
|---|---|---|---|---|---|---|---|
| **Demographic characteristics** | | | | | | | |
| Age at enrollment (years), median (IQR) | | 57.0 (47.0–66.0) | 57.0 (45.0–66.0) | 61.0 (50.5–69.0) | 54.5 (46.0–62.0) | 60.0 (52.0–71.0) | 0.006 |
| Sex at birth, no. (%) | Male | 360 (61) | 238 (66) | 49 (53) | 42 (51) | 31 (53) | 0.009 |
| | Female | 230 (39) | 120 (34) | 43 (47) | 40 (49) | 27 (47) | |
| Race, no. (%)[a] | White | 289 (49) | 175 (49) | 55 (60) | 27 (33) | 32 (55) | 0.011 |
| | Black | 131 (22) | 82 (23) | 16 (17) | 20 (24) | 13 (22) | |
| Ethnicity, no. (%)[b] | Non-Hispanic | 384 (65) | 233 (65) | 57 (62) | 49 (60) | 45 (78) | 0.313 |
| | Hispanic | 189 (32) | 113 (32) | 32 (35) | 32 (39) | 12 (21) | |
| **Comorbidities, no. (%)** | | | | | | | |
| | Hypertension | 327 (55) | 183 (51) | 56 (61) | 50 (61) | 38 (66) | 0.066 |
| | Diabetes | 200 (34) | 108 (30) | 37 (40) | 33 (40) | 22 (38) | 0.124 |
| | Chronic respiratory (not asthma) | 104 (18) | 48 (13) | 27 (29) | 14 (17) | 15 (26) | 0.001 |
| | Asthma | 101 (17) | 56 (16) | 20 (22) | 16 (20) | 9 (16) | 0.499 |
| | Chronic cardiac disease | 142 (24) | 70 (20) | 29 (32) | 22 (27) | 21 (36) | 0.008 |
| | Chronic kidney disease | 69 (12) | 44 (12) | 9 (10) | 12 (15) | 4 (7) | 0.488 |
| | Malignant neoplasm | 49 (8) | 32 (9) | 9 (10) | 5 (6) | 3 (5) | 0.631 |
| | Chronic neurologic disorder | 61 (10) | 22 (6) | 14 (15) | 9 (11) | 16 (28) | <0.001 |
| | Liver disease | 28 (5) | 15 (4) | 6 (7) | 4 (5) | 3 (5) | 0.822 |
| | History of SOT or BMT | 31 (5) | 16 (4) | 4 (4) | 7 (9) | 4 (7) | 0.443 |
| | Current or former smoking and/or vaping | 182 (31) | 101 (28) | 33 (36) | 24 (29) | 24 (41) | 0.147 |
| | Substance use (drugs, alcohol, and/or cannabis) | 40 (7) | 22 (6) | 5 (5) | 5 (6) | 8 (14) | 0.167 |
| **BMI category in Kg/m², no. (%)[b]** | Normal weight | 65 (11) | 41 (11) | 5 (5) | 12 (15) | 7 (12) | 0.021 |
| | Overweight (25.1–29.9) | 162 (27) | 108 (30) | 19 (21) | 20 (24) | 15 (26) | |
| | Class 1–2 Obesity (30–39.9) | 256 (43) | 151 (42) | 43 (47) | 36 (44) | 26 (45) | |
| | Class 3 Obesity (40+) | 90 (15) | 43 (12) | 25 (27) | 13 (16) | 9 (16) | |

**Table 1 (continued) | Baseline characteristics of the survey respondent cohort**

| | | Overall (n = 590) | Cluster MIN (n = 358, 60.7%) | Cluster PHY (n = 92, 15.6%) | Cluster COG (n = 82, 13.9%) | Cluster MLT (n = 58, 9.8%) | Overall p value |
|---|---|---|---|---|---|---|---|
| Number of comorbidities, median (IQR) | (n = 590) | 3.0 (2.0–40) | 3.0 (1.0–4.0) | 4.0 (2.5–5.0) | 3.0 (2.0–5.0) | 3.0 (2.0–6.0) | <0.001 |
| | None | 35 (6) | 30 (8) | 0 (0) | 3 (4) | 2 (3) | 0.006 |
| | 1 | 92 (16) | 68 (19) | 9 (10) | 11 (13) | 4 (7) | |
| | 2 | 116 (20) | 75 (21) | 14 (15) | 13 (16) | 14 (24) | |
| | 3 | 118 (20) | 65 (18) | 21 (23) | 22 (27) | 10 (17) | |
| | 4 | 83 (14) | 45 (13) | 17 (18) | 12 (15) | 9 (16) | |
| | 5 or more | 146 (25) | 75 (21) | 31 (34) | 21 (26) | 19 (33) | |
| **Baseline visit** | | | | | | | |
| Infiltrates on chest X-ray or chest tomography | No infiltrates | 137 (25) | 80 (24) | 27 (30) | 17 (23) | 13 (23) | 0.645 |
| | Unilateral infiltrates | 50 (9) | 32 (9) | 8 (9) | 3 (4) | 7 (13) | |
| | Bilateral infiltrates | 369 (66) | 225 (66) | 53 (60) | 55 (73) | 36 (64) | |
| Level of respiratory support | Mechanically ventilated or ECMO | 41 (7) | 22 (6) | 6 (7) | 9 (11) | 4 (7) | 0.075 |
| | Non-invasive ventilation or high-flow oxygen | 124 (21) | 86 (24) | 12 (13) | 19 (23) | 7 (12) | |
| | Supplemental oxygen (not high flow) | 284 (48) | 173 (48) | 52 (57) | 32 (39) | 27 (47) | |
| | None | 141 (24) | 77 (22) | 22 (24) | 22 (27) | 20 (34) | |
| SpO2/FiO2 ratio category at lowest sat[c] | 235 or lower | 100 (17) | 67 (19) | 9 (10) | 16 (20) | 8 (14) | 0.537 |
| | 236–315 | 101 (17) | 58 (16) | 19 (21) | 16 (20) | 8 (14) | |
| | 315 or higher | 355 (60) | 214 (60) | 56 (61) | 46 (56) | 39 (67) | |
| SOFA score, median (IQR) | (n = 590) | 0.0 (0–2.0) | 0.0 (0–2.0) | 0.0 (0–2.0) | 0.5 (0–3.0) | 0.0 (0.0–2.0) | 0.435 |
| **Baseline labs** | | | | | | | |
| Lymphocyte count (1000 s/microliter), median (IQR)[d] | (n = 495) | 1.1 (0.7–1.7) | 1.0 (0.7–1.6) | 1.2 (0.7–1.8) | 1.1 (0.5–1.6) | 1.2 (0.7–1.9) | 0.371 |
| Platelet count (1000 s/microliter), median (IQR)[c] | (n = 560) | 238.5 (178–303) | 241.0 (189–313) | 218.5 (149–282) | 247.0 (180–292) | 222.0 (166–282) | 0.017 |
| ALT (Units/L), median (IQR)[e] | (n = 515) | 33.0 (22–57) | 34.0 (23–59) | 30.0 (19–48) | 33.5 (21–56) | 31.0 (22–50) | 0.378 |
| Creatinine (mg/dL), median (IQR)[b] | (n = 568) | 0.9 (0.7–1.1) | 0.9 (0.7–1.1) | 0.9 (0.7–1.1) | 0.8 (0.7–1.1) | 0.8 (0.7–1.1) | 0.991 |
| CRP (mg/L), median (IQR)[f] | (n = 418) | 14.5 (6.0–74.7) | 14.5 (6.7–76.5) | 12.4 (5.0–82.2) | 26.0 (6.8–93.0) | 9.0 (4.5–60.4) | 0.048 |
| D-dimer (mg/L), median (IQR)[f] | (n = 416) | 0.7 (0.5–1.3) | 0.8 (0.5–1.4) | 0.5 (0.4–0.8) | 0.9 (0.4–1.8) | 0.7 (0.5–1.3) | 0.008 |
| Troponin (ng/mL), median (IQR)[g] | (n = 199) | 0.0 (0.0–0.06) | 0.0 (0.01–0.06) | 0.0 (0.01–2.9) | 0.0 (0.01–0.05) | 0.0 (0.02–0.04) | 0.252 |
| **Severity & utilization during acute hospitalization** | | | | | | | |
| Length of stay (days), median (IQR) | (n = 590) | 5.0 (4.0–10.0) | 5.0 (3.0–9.0) | 5.0 (3.0–9.0) | 6.0 (5.0–12.0) | 6.0 (5.0–12.0) | 0.021 |
| ICU at any time during acute hospitalization | | 159 (27) | 92 (26) | 23 (25) | 25 (30) | 19 (33) | 0.577 |
| Acute trajectory group | TG1y | 145 (25) | 97 (27) | 24 (26) | 15 (18) | 9 (16) | 0.327 |

**Table 1 (continued) | Baseline characteristics of the survey respondent cohort**

| | | Overall (n = 590) | Cluster MIN (n = 358, 60.7%) | Cluster PHY (n = 92, 15.6%) | Cluster COG (n = 82, 13.9%) | Cluster MLT (n = 58, 9.8%) | Overall p value |
|---|---|---|---|---|---|---|---|
| | TG2 | 195 (33) | 109 (30) | 27 (29) | 33 (40) | 26 (45) | |
| | TG3 | 151 (26) | 93 (26) | 26 (28) | 18 (22) | 14 (24) | |
| | TG4 | 99 (17) | 59 (16) | 15 (16) | 16 (20) | 9 (16) | |
| Acute complications reported | | | | | | | |
| Any complications | | 490 (83) | 308 (86) | 72 (78) | 66 (80) | 44 (76) | 0.099 |
| Number of complications, median (IQR) | (n = 590) | 2.0 (1.0–3.0) | 2.0 (1.0–3.0) | 2.0 (1.0–3.0) | 2.0 (1.0–3.0) | 2.0 (1.0–3.0) | 0.95 |
| Acute renal injury/failure | | 94 (16) | 55 (15) | 20 (22) | 13 (16) | 6 (10) | 0.289 |
| Liver dysfunction/failure | | 68 (12) | 47 (13) | 7 (8) | 5 (6) | 9 (16) | 0.135 |
| Anemia | | 62 (11) | 33 (9) | 10 (11) | 16 (20) | 3 (5) | 0.024 |
| Shock (use of vasopressors) | | 45 (8) | 20 (6) | 8 (9) | 8 (10) | 9 (16) | 0.048 |
| Bacteremia | | 39 (7) | 20 (6) | 8 (9) | 6 (7) | 5 (9) | 0.637 |
| Atrial fibrillation | | 32 (5) | 16 (4) | 6 (7) | 6 (7) | 4 (7) | 0.644 |
| Acute venous thromboembolism | | 29 (5) | 16 (4) | 5 (5) | 6 (7) | 2 (3) | 0.686 |
| Congestive heart failure (CHF)/cardiomyopathy | | 26 (4) | 14 (4) | 6 (7) | 4 (5) | 2 (3) | 0.716 |
| Hyperglycemia | | 24 (4) | 14 (4) | 4 (4) | 3 (4) | 3 (5) | 0.968 |
| Medications during acute hospitalization | | | | | | | |
| Steroids | | 400 (68) | 251 (70) | 63 (68) | 55 (67) | 31 (53) | 0.094 |
| Remdesivir | | 377 (64) | 236 (66) | 60 (65) | 47 (57) | 34 (59) | 0.4 |
| Convalescent symptoms | | | | | | | |
| Any symptom reported during convalescence | | 305 (52) | 149 (42) | 58 (64) | 55 (68) | 43 (74) | <0.001 |
| Any upper respiratory (sore throat, conjunctivitis) | | 106 (18) | 38 (11) | 18 (20) | 25 (31) | 25 (43) | <0.001 |
| | Conjunctivitis/red eyes | 79 (13) | 26 (7) | 14 (16) | 17 (21) | 22 (38) | <0.001 |
| | Sore throat | 43 (7) | 14 (4) | 7 (8) | 10 (12) | 12 (21) | <0.001 |
| Any cardiopulmonary (cough, dyspnea) | | 204 (35) | 92 (26) | 37 (41) | 37 (46) | 38 (66) | <0.001 |
| | Dyspnea | 171 (29) | 73 (20) | 31 (34) | 33 (41) | 34 (59) | <0.001 |
| | Cough | 116 (20) | 44 (12) | 26 (29) | 21 (26) | 25 (43) | <0.001 |
| Any systemic (fever, fatigue, myalgia, chills) | | 173 (30) | 65 (18) | 35 (39) | 38 (47) | 35 (60) | <0.001 |
| | Fatigue | 107 (18) | 33 (9) | 17 (19) | 29 (36) | 28 (48) | <0.001 |
| | Myalgia | 121 (21) | 39 (11) | 24 (27) | 29 (36) | 29 (50) | <0.001 |
| | Fever | 27 (5) | 9 (3) | 5 (6) | 8 (10) | 5 (9) | 0.012 |
| | Chills | 15 (3) | 6 (2) | 4 (4) | 4 (5) | 1 (2) | 0.223 |

**Table 1 (continued) | Baseline characteristics of the survey respondent cohort**

| | | Overall (n = 590) | Cluster MIN (n = 358, 60.7%) | Cluster PHY (n = 92, 15.6%) | Cluster COG (n = 82, 13.9%) | Cluster MLT (n = 58, 9.8%) | Overall p value |
|---|---|---|---|---|---|---|---|
| Any neurologic (headache, anosmia) | | 162 (28) | 64 (18) | 27 (30) | 39 (48) | 32 (55) | <0.001 |
| | Head-ache | 112 (19) | 40 (11) | 20 (22) | 27 (33) | 25 (43) | <0.001 |
| | Anos-mia | 83 (14) | 29 (8) | 13 (14) | 21 (26) | 20 (34) | <0.001 |
| Any gastrointestinal (nausea/vomiting) | | 41 (7) | 11 (3) | 11 (12) | 9 (11) | 10 (17) | <0.001 |

Cluster MIN: minimal deficit, cluster PHY: deficit, physical predominant, cluster COG: deficit, mental/cognitive predominant and cluster MLT: deficit, multidomain.
SOT solid organ transplantation, BMT bone marrow transplantation, ECMO extracorporeal membrane oxygenation, SpO2/FiO2 saturation of oxygen/fraction of inspired oxygen, SOFA sequential organ failure assessment score, ICU intensive care unit, ALT alanine transaminase; CRP: C-reactive protein.
Two-sided p values without adjustment for multiple comparisons are reported from Wilcoxon rank-sum test for continuous age, $x^2$ test for categorical variables.
[a]28% other/declined/unknown/missing.
[b]Less than 5% missing data.
[c]Less than 10% missing data.
[d]Less than 20% missing data.
[e]Less than 15% missing data.
[f]Less than 30% missing data.
[g]66% did not have troponin level on admission.

received less oxygen supplementation OR = 0.54 (95% CI: 0.34, 0.87) and had longer acute hospitalization (OR per week 1.44; 95% CI 1.19–1.75) relative to those in the MIN cluster (Fig. 2).

**Laboratory assays by PRO cluster**

N1 gene SARS-CoV-2 PCR cycle threshold (Ct) values up to 28 days since admission differed significantly among the four PRO clusters (Fig. 3 and Fig. 4S). Specifically, the viral RNA levels from the respiratory tract were significantly higher (lower Ct values) throughout those 28 days in participants within both the PHY and MLT clusters compared the MIN cluster in the longitudinal GAM model (Fig. 4S, adj. $p = 0.015$). There was no difference in Ct values between clusters with MIN and COG deficits. Viral levels declined over time in all four PRO clusters and the rate of viral clearance did not differ significantly among the PRO clusters (Fig. 4S). Similar trends were also observed with N2 SARS-CoV-2 PCR Ct values (Fig. 4S, adj. $p = 0.013$).

Acute phase anti-SARS-CoV-2 RBD IgG and Spike IgG titers, were also significantly associated with PRO clusters (Fig. 3 and Fig. 4S). Both anti-RBD IgG and anti-Spike IgG levels were significantly lower in the MLT compared to the MIN and COG clusters by the GAM model (Fig. 4S, adj. $p = 0.023$) and in the PHY compared to the MIN and COG clusters by the GLM model (Figs. 3 and 4S, adj. $p = 0.014$). There was a swift rise in antibody (Ab) levels during first 7 days followed by a modest increase between 7 and 20 days, reaching a plateau at day 28 in all PRO clusters. We also observed significantly faster rise in Ab levels between 7 and 20 days for clusters PHY and MLT compared to the other 2 clusters (Fig. 4S; shape, anti-RBD IgG adj. $p = 0.005$ and anti-Spike IgG adj. $p = 0.0017$). The ratio of anti-RBD IgG to N1 Ct followed similar patterns (Fig. 4S).

We next considered whether circulating leukocyte subset frequencies during the acute phase correlated with PRO clusters and carried out deep immunophenotyping using CyTOF to quantify cell frequencies in whole blood. Our analysis of cell subsets revealed that in the first 28 days of hospitalization, differences in the frequency of circulating B lymphocytes were significantly associated with PRO clusters (Fig. 4, adj. $p = 0.0191$, Participants with MLT deficits (adj. $p = 0.0005$) and COG deficits (adj. $p = 0.025$) had a significantly lower frequency of circulating B cells compared to those with MIN deficits (Fig. 4). Notably, within B cells, participants with MLT deficits showed a lower frequency of naïve B cells compared to MIN cluster, although the difference did not remain significant after multiple-comparison correction for all B cell subsets (Fig. 5S). Other immune cell subtypes were not significantly associated with PRO cluster. The ratios of SARS-CoV-2 PCR Ct values and antibody titers to B cell numbers are shown in Fig. 6S, indicating that the lower frequency of circulating B cells may be relevant to outcome in the context of distinct kinetics of Ab production and viral clearance.

Autoantibodies with blocking activity against type I interferons (alpha, beta, and omega) were observed at the onset of hospitalization in 4.3% participants (24 out of 563) across four PRO clusters (Tanle 2S). Anti-interferon autoantibodies were detected in 5.3% of males and 2.7% female, ($P = 0.14$) and were more common in individuals older than 65 years of age (8.9% older vs 2.8% younger, $P = 0.039$). We observed a proportionately larger fraction of individuals with positive autoantibodies against type I IFNs from PHY and MLT PRO clusters (PHY cluster = 6.9%, MLT cluster = 7.3%) compared to the other 2 clusters (MIN cluster = 3.2%, COG cluster = 3.8%) ($p = 0.2$). In a matched case-control (1:3) based on age and sex comparison, individuals with IFN autoantibodies had significantly higher viral loads than individuals without IFN autoantibodies (Fig. 7S, N1 Ct $P = 0.012$ and N2 Ct $P = 0.006$). We did not see significant differences in Ab titers against either SARS-CoV-2 RBD IgG or spike IgG levels between these matched groups.

Using a panel of 92 inflammatory markers, we analyzed serum samples using a proximity extension assay (Olink) across the period of

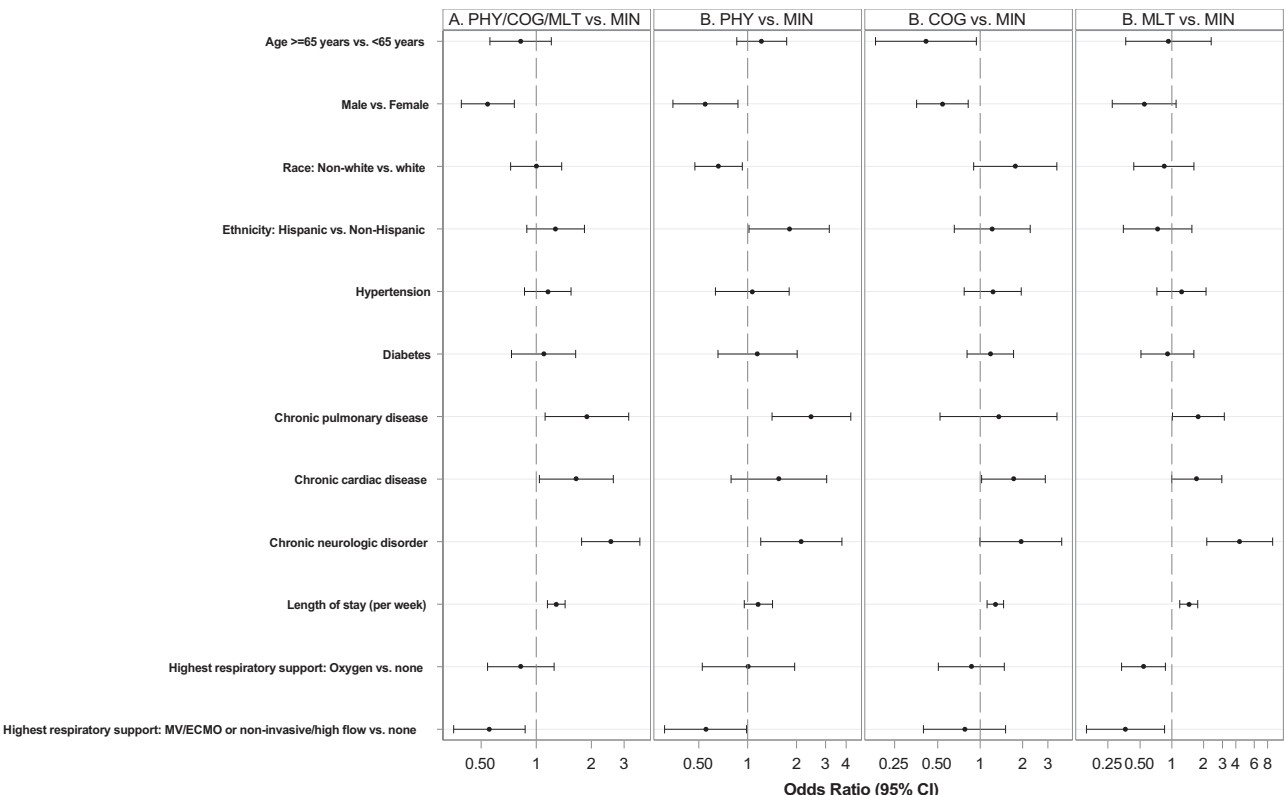

**Fig. 2 | Forest Plot showing adjusted odds ratios (ORs) for factors associated with patient-reported outcome (PRO) clusters with more deficits compared to minimal deficit, using multivariable multinomial logistic regression (N = 590). A** Comparison of PRO clusters PHY, COG, and MLT with PRO cluster MIN. **B** Comparison of PRO clusters PHY and MIN (left); clusters COG and MIN (middle); clusters MLT and MIN (right). MIN minimal deficit, PHY physical predominant deficit, COG mental/cognitive predominant deficit, MLT multidomain deficit, MV Mechanical ventilation, ECMO Extracorporeal membrane oxygenation.

acute illness; one protein, fibroblast growth factor 21 (FGF21), was significantly elevated in the COG cluster (adj. $p = 0.0025$) as well as the MLT cluster (adj. $p = 0.000033$), relative to the MIN cluster. The highest mean FGF21 values were in the MLT cluster (Fig. 5, Fig. 8S).

Analyzing 658 serum metabolites, 27 modules were identified from a weighted gene correlation network analysis (WGCNA) which corresponded to each metabolite feature. We observed a significant difference in shape (referred to as the smoothing term in the gamm4 documentation) for methylhistidine metabolism (global metabolomics module 3) and acylcarnitine metabolism (global metabolomics module 18) among the PRO clusters (Fig. 9SA, SB). Notably, significantly lower levels of metabolites related to methylhistidine metabolism were observed for participants in the PHY and the MLT clusters, compared to the MIN cluster (shape adj. $p = 0.049$). Further, significantly higher levels of metabolites related to acylcarnitine metabolism were observed for participants in the PHY cluster, compared to the MIN cluster (adj. $p = 0.049$).

## Discussion

In this large prospective study that followed participants from the time of acute COVID-19 hospitalization, more than half of the participants hospitalized with COVID-19 had persistent symptoms lasting 3 or more months after discharge, consistent with other studies[9]. The clustering of Patient-Reported Outcomes (PROs) by predominant deficit (physical predominant, mental/cognitive predominant, and multidomain deficits) supports PASC as a heterogeneous clinical entity with distinct sub-phenotypes associated with unique perturbations of the immune system in the acute phase of the illness[10,11]. This cluster-based analysis also revealed a specific participant phenotype associated with persistent mental and cognitive impairments that were distinct from phenotypes associated with broader physical dysfunction. These findings

suggest tailored approaches will be needed in the management of PASC[12-14].

Overall demographic and clinical risk factors for PASC in our cohort include female sex, comorbidities[13,15,16] such as chronic heart, lung, or neurologic disease, as well as longer length of hospital stay[17]. The biological basis for why females may be more susceptible to PASC than males has yet to be defined, though several models have been proposed[18,19]. One potential mechanism is autoimmunity, though we did not find any difference between males and females in autoantibodies against type I interferons. Hormonal factors may also play a role in perpetuating the hyperinflammatory status of the acute disease phase even after initial recovery[20]. While certain comorbidities are associated with PASC (e.g., chronic pulmonary diseases in the physical predominant deficit cluster), it is unclear based on our data if some of the PASC disease burden could be misattributed to COVID-19 or that COVID-19 accentuates these pre-existing conditions. Consistent with prior reports in non-hospitalized patients, we found no association between PASC and acute COVID-19 disease severity[21-24], but a longer length of stay was associated with all PRO deficit clusters when compared to those with minimal functional deficits, similar to findings from a population-based cohort study[25]. Interestingly, our results suggest that supportive interventions such as oxygen therapy may be associated with a lower likelihood of being in the multidomain deficit PRO cluster, supporting the notion that early acuity-based interventions may positively influence clinical outcomes[26].

Our data demonstrate that higher SARS-CoV-2 viral burden and lower Ab titers during the acute phase are associated with both the physical predominant deficit as well as the multidomain deficit PRO clusters. Of note, these virologic and serologic findings do not distinguish participants with minimal deficits from those in the mental/

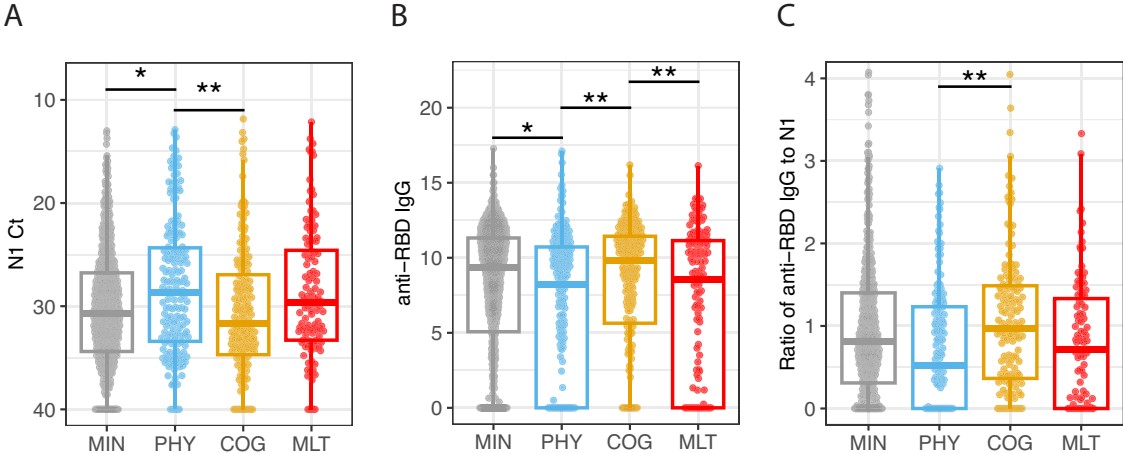

**Fig. 3 | SARS-CoV-2 viral RNA levels and antibody responses. A** N1 Ct values: shown are SARS-CoV-2 N1 gene PCR cycle threshold (Ct) values (viral loads) measured from samples collected during the first 28 days of hospital admission by four PRO clusters, minimal deficit (MIN, $n = 657$), physical predominant (PHY, $n = 174$), deficit, mental/cognitive predominant (COG, $n = 172$ and deficit, multidomain (MLT, $n = 112$). Shown are median values (horizontal lines), interquartile ranges (boxes), and 1.5 IQR (whiskers), as well as all individual points. Because lower Ct values indicate higher viral loads, the y axis is reversed. The viral loads were significantly (adj. $p = 0.03$) associated with the PRO clusters. **B** anti-RBD IgG values: Shown are anti-RBD IgG values measured from samples collected during the first 28 days of hospital admission by four PRO clusters, minimal deficit (MIN, $n = 907$), physical predominant (PHY, $n = 221$), deficit, mental/cognitive predominant (COG, $n = 230$) and deficit, multidomain (MLT, $n = 149$). Shown are median values of area under the curve (AUC) (horizontal lines), interquartile ranges (boxes), and 1.5 IQR (whiskers), as well as all individual points. The titers were significantly (adj. $p = 0.014$) associated with the PRO clusters. **C** Ratio of anti-RBD IgG to N1 values:

shown are scaled ratio of anti-RBD IgG to SARS-CoV-2 viral loads (N1 gene) values from samples collected during the first 28 days of hospital admission by four PRO clusters, minimal deficit (MIN, $n = 560$), physical predominant (PHY, $n = 156$), deficit, mental/cognitive predominant (COG, $n = 141$) and deficit, multidomain (MLT, $n = 99$). Shown are median values (horizontal lines), interquartile ranges (boxes), and 1.5 IQR (whiskers), as well as all individual points. The ratio of titers to viral loads was also significantly (adj. $p = 0.05$) associated with the PRO clusters. The four PRO clusters are the following in gray: minimal deficit (MIN), in blue: deficit, physical predominant (PHY), in yellow: deficit, mental/cognitive predominant (COG), and in red: deficit, multidomain (MLT). The lines and asterisks on top of the figure denote pairwise statistical significance, *$p < 0.05$, **$p < 0.01$, ***$p < 0.001$. Statistical differences were determined from generalized linear mixed effects models adjusting for age, sex, participant, and enrollment site. P values were adjusted using the Benjamini-Hochberg method to account for multiple comparisons. See Methods for more details.

cognitive predominant deficit PRO cluster, suggesting that different factors could lead to the development of this particular cluster. The described calculated ratio (IgG/Ct value) is unique not only in the acute phase to determine the trajectory of acute disease course but also associates with PRO clustering in the convalescent phase[8], and thus may represent a practical approach for patient risk stratification for both early mortality and subsequent morbidity from PASC. Our data confirm findings from other studies suggesting that PASC is associated with initial high SARS-CoV-2 RNA levels[27] and a suboptimal serological response[28–30] consistent with a reduced number of circulating B cells mostly in the multidomain deficit cluster. The therapeutic and protective effects of immunoglobulins were the basis for the pre-Omicron use of convalescent plasma and monoclonal antibodies (mAbs) and are the basis for the continued use of vaccination in the treatment and prevention of COVID-19. Recent observations of PASC resolution after SARS-CoV-2 vaccination raise the possibility of depletion of persisting viral reservoirs[31]. Since none of our study participants (enrolled in 2020 and early 2021) were vaccinated prior to their illness and only 6% received convalescent plasma, it was not possible to determine the impact of early Ab therapy or vaccination on the likelihood of PASC. Regarding other COVID-19-directed interventions, use of remdesivir and steroids in the inpatient period was not associated with a decrease in PASC prevalence. However, a recent study reported a protective effect against PASC when another antiviral, nirmatrelvir/ritonavir, was used in the acute phase of COVID-19, consistent with our finding of high viral load being associated with the development of PASC[32,33]. PASC has been associated with the detection of different auto-antibodies early in the disease course[6,27,34–36]. In our study, the presence of IFN-specific autoantibodies is associated with viral load and severity of acute COVID[37]. Autoantibodies neutralizing IFNα, IFNβ, and/or IFNω result in a persistent dampening of IFN responses, likely leading to

insufficient viral clearance (as seen in our study in our case-control subanalysis) and tissue damage.

While numerous cytokines have been associated with PASC in other studies, in this study, we found that only fibroblast growth factor 21 (FGF21) was significantly associated with the PRO clusters[38,39]. FGF21 is a cytokine known to regulate systemic glucose and lipid metabolism that is secreted from muscle in response to stress[40] or even infection, particularly mitochondrial myopathy[41] supporting a potential catabolic role of FGF21 on human muscle health[42]. In our metabolomics data, we found a significant association between global metabolomics module 3 (methylhistidine metabolism) and the PRO clusters, consistent with the inverse relationship between 3-methylhistidine (3MH) and FGF21, thought to be mediated by insulin sensitivity[43]. This observation is also consistent with the relationship between 3-methylhistidine and muscle cells, in which 3MH is a potential biomarker for muscle atrophy and skeletal muscle toxicity. Whether the association we observed with FGF21 reflects underlying mitochondrial dysfunction as the pathobiological basis for PASC is unclear and deserves further investigation[43]. Elevation of plasma FGF21 has also been noted in patients with myalgic encephalomyelitis/chronic fatigue syndrome (ME/CFS), an entity with clinical features that overlaps with PASC including fatigue, post-exertional malaise, sleep disturbance, and brain fog[44]. Interestingly, acylcarnitines also noted to be significant comparing PHY and MIN cluster, are known to have an essential role in metabolism and breaking down fatty acids for energy production[45,46]. In addition, acyl carnitine substrates have been investigated as surrogate biomarkers for ME/CFS[47,48] and impaired metabolic health[49], suggesting its possible role in reduced physical function after COVID-19 hospitalization.

Overall, participants in the MLT cluster with the most severe functional deficits during the year following hospitalization for COVID-

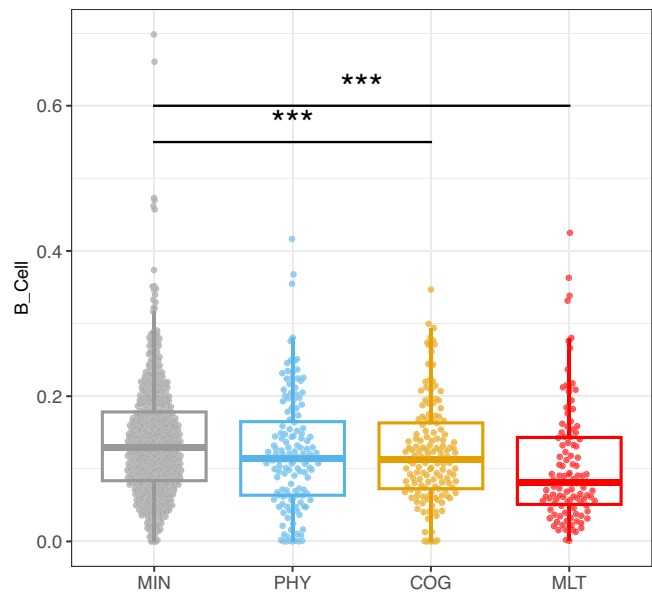

**Fig. 4 | B cell to non-granulocyte frequency.** Shown are B cell to non-granulocyte frequency values from samples collected during the first 28 days of hospital admission by four PRO clusters, minimal deficit (MIN, $n = 584$), physical predominant (PHY, $n = 140$), deficit, mental/cognitive predominant (COG, $n = 145$) and deficit, multidomain (MLT, $n = 107$). Shown are median values (horizontal lines), interquartile ranges (boxes), and 1.5 IQR (whiskers), as well as all individual points. The repeated-measurement model identified significant differences of B cell to non-granulocyte frequency in association with convalescent clusters (adj. $p = 0.0191$). The 4 clusters are the following in gray: minimal deficit (MIN), in blue: deficit, physical predominant (PHY), in yellow: deficit, mental/cognitive predominant (COG), and in red: deficit, multidomain (MLT). The lines and asterisks on top of the figure denote pairwise statistical significance, *$p < 0.05$, **$p < 0.01$, ***$p < 0.001$. Statistical differences were determined from generalized linear mixed effects models adjusting for age, sex, participant, and enrollment site. $P$ values were adjusted using the Benjamini–Hochberg method to account for multiple comparisons. See Methods for more details.

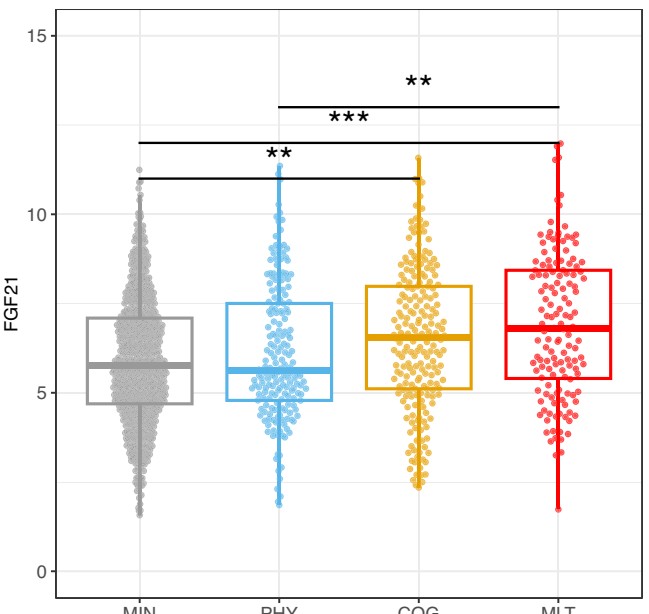

**Fig. 5 | Circulating fibroblast growth factor 21 expression.** Circulating fibroblast growth factor 21 (FGF21) NPX (Normalized protein expression): Shown are FGF21 NPX values from samples collected during the first 28 days of hospital admission by four PRO clusters, minimal deficit (MIN, $n = 716$), physical predominant (PHY, $n = 189$), deficit, mental/cognitive predominant (COG, $n = 210$) and deficit, multidomain (MLT, $n = 139$). Shown are median values (horizontal lines), interquartile ranges (boxes), and 1.5 IQR (whiskers), as well as all individual points. The generalized additive model (GAM) identified a significant difference in FGF21 expression level in association with convalescent cluster groups (adj. $p = 0.0135$). The four clusters are the following in gray: minimal deficit (MIN), in blue: deficit, physical predominant (PHY), in yellow: deficit, mental/cognitive predominant (COG), and in red: deficit, multidomain (MLT). Statistical differences were determined from generalized linear mixed effects models adjusting for age, sex, participant, and enrollment site. $P$ values were adjusted using the Benjamini–Hochberg method to account for multiple comparisons. See Methods for more details.

19 had a suboptimal serological response likely driven by low circulating B cells potentially leading to high viral replication in the acute phase of the disease. In a case-control analysis, the highest percentage of autoantibodies against interferon (IFN) was noted in the MLT cluster, and the presence of these autoantibodies correlated with a high viral load. In addition, we noted an elevation in FGF21 levels in the MLT cluster similar to the COG cluster indicating possible muscle injury/stress that could explain common PASC symptoms such as fatigue and malaise. Interestingly, the COG cluster was not associated with a high viral load nor suboptimal serological response, suggesting the possibility that early immunologic parameters associated with mental and cognitive deficits following COVID-19 may be distinct from those leading to other functional limitations.

This study has several strengths, including enrollment of a diverse population from a wide variety of geographically dispersed hospitals and detailed clinical and biological phenotyping, as well as several limitations. Due to the timing of our recruitment window, very few study participants were infected with variants of concern or variants of interest or vaccinated prior to hospital admission[8]; thus, symptom persistence in our cohort may not be representative of patients infected with more recent emerging SARS-CoV-2 variants or with breakthrough infections[50], but does provide an important characterization of post-acute disease follow-up in a virus-naïve population.

Along the same lines, certain symptoms that are now frequently linked to PASC (e.g., 'brain fog', sleep disturbance, dysautonomia) were not recorded as the surveys were designed prospectively early in the pandemic (March 2020) when PASC had not yet been reported. However, our use of standardized PROs that targeted cognitive, mental, and psychosocial functions enabled the identification of a specific cluster with predominant mental and cognitive deficits. Although symptoms at hospital admission were captured, pre-COVID symptomatology was not recorded, limiting estimation of the proportion of persistent symptoms directly attributable to PASC versus part of a pre-existing co-morbid condition. In addition, we did not attempt to identify alternative causes of persistent or new symptoms. However, PRO measures were chosen to attempt to mitigate this limitation by including a comparison to pre-illness baseline or some other appropriate recall period when possible. Our incidence of new onset or worsening health-related quality measures is in line with other published studies with 31% impairment in one study[51], and 15.4% with poor physical component, and 32.6% with poor mental component in another study[52].

Other investigators observed that certain occupations and socioeconomic status are associated with PASC[15,16]; we were unable to assess such associations due to a lack of occupational and socioeconomic data in our study.

Exclusion of non-hospitalized patients also affects the generalizability of our findings to patients with COVID-19 not requiring hospitalization. Also, our study did not include control groups: (1) who did not have COVID, (2) hospitalized for a non-COVID-19 respiratory viral infection, and/or (3) hospitalized for elective procedures where the length of stay is similar to COVID-19.

T-cell dysfunction has been described in PASC[53], but our study did not include this assessment. We also did not fully explore other

autoantibodies beyond those against type I IFNs (e.g., Ro/SS-A, La/SS-B, U1-snRNP, Jo-1, and P1[54] to β2 adrenoceptor, muscarinic M2 receptor, angiotensin II AT1 receptor, and angiotensin 1–7 MAS receptor)[6]. Similarly, EBV reactivation has been reported as potentially associated with PASC and was not investigated here.

We did not have an independent validation cohort, however, enrollment at multiple sites may have decreased selection biases.

While we largely report immunologic results observed in the analysis of blood, we also evaluated the acute immune response in the upper airway in non-intubated participants and in the lower airway in participants receiving mechanical ventilation; studies that did not detect significant associations with convalescent phenotypes. Acknowledging the broad organotropism of SARS-CoV-2, future research should explore other compartments serving as potential viral reservoirs[55].

In addition to preventing and treating acute infections, there is a dire need to better understand and develop treatments for individuals with PASC. Our study represents a large multi-site prospective cohort with extensive clinical data capture and 12 months of follow-up after discharge, as well as intensive immunophenotyping efforts that employed a variety of innovative assays with rigorous data management and a standardized analysis pipeline. Our findings suggest that a functional antiviral Ab immune response contributes to viral clearance and may decrease the occurrence of PASC presenting as significant physical and multidomain deficits. Our results also highlight the benefit of measuring immune responses during the acute phase for the early identification of patients at high risk for PASC, which may facilitate testing and monitoring of targeted PASC prevention and treatment.

## Methods

### Ethics
NIAID staff conferred with the Department of Health and Human Services Office for Human Research Protections (OHRP) regarding the potential applicability of the public health surveillance exception [45CFR46.102(l) (2)] to the IMPACC study protocol. OHRP concurred that the study satisfied criteria for the public health surveillance exception, and the IMPACC study team sent the study protocol, and participant information sheet for review and assessment to institutional review boards (IRBs) at participating institutions. Twelve institutions elected to conduct the study as public health surveillance, while 3 sites with prior IRB-approved biobanking protocols elected to integrate and conduct IMPACC under their institutional protocols (University of Texas at Austin, IRB 2020-04-0117; University of California San Francisco, IRB 20-30497; Case Western Reserve University, IRB STUDY20200573) with informed consent requirements. Participants enrolled under the public health surveillance exclusion were provided information sheets describing the study, samples to be collected, and plans for data de-identification and use. Those who requested not to participate after reviewing the information sheet were not enrolled. In addition, participants did not receive compensation for study participation while inpatient, and subsequently were offered compensation during outpatient follow-ups.

### Study design and setting
The study followed the Strengthening the Reporting of Observational Studies in Epidemiology (STROBE) guidelines for reporting observational studies[56]. This study was registered at clinicaltrials.gov (NCT0438777).

### Study participants
Patients 18 years and older admitted to 20 US hospitals (affiliated with 15 academic institutions) between May 2020 and March 2021 were enrolled within 72 h of hospital admission for COVID-19 infection. Only confirmed positive SARS-CoV-2 PCR and symptomatic cases attributable to COVID-19 infection were followed longitudinally[57]. Participants were provided compensation on an outpatient basis.

### Data collection, study variables, and biologic samples
Specific data elements were acquired via a review of electronic medical records during the inpatient period[8]. The study was designed to enroll participants of both sexes, and sex at birth was collected based on self-report or caregiver report. Length of hospital stay, complications, mortality, and other protocol-defined outcomes were assessed over 28 days. In addition, self-reported symptoms, reinfections, SARS-CoV-2 vaccination, rehospitalizations, and standardized patient-reported outcome surveys were assessed quarterly for the duration of the study up to 12 months after initial hospital discharge.

Biologic samples collected consisted of blood and mid-turbinate nasal swabs (self, or staff collected). The timepoints were as follows: enrollment (Day 1), and Days 4, 7, 14, 21, and 28 post hospital admission (and if feasible, for discharged participants, Days 14 and 28).

Patient-reported data was collected using a comprehensive digital remote monitoring tool, in the form of a mobile application developed by *My Own Med, Inc.* Along with the mobile application, an administrative portal was developed to collect information by study personnel during site visits or via telephone interviews with a study coordinator to ensure real-time electronic data capture.

The surveys administered at these remote visits:[58]
- Upper respiratory symptoms: sore throat, conjunctivitis/red eyes
- Cardiopulmonary symptoms: shortness of breath (dyspnea), cough
- Systemic symptoms: fever, chills, fatigue/malaise, muscle aches (myalgia)
- Neurologic symptoms: loss of smell/taste (anosmia/ageusia), headache
- Gastrointestinal symptoms: nausea/vomiting

In addition, the functional assessments of general health and the evaluation of deficits in specific health domains were conducted using validated Patient-Reported Outcome (PRO) measures, including:
- EQ-5D-5L, a standardized, self-administered instrument that describes and quantifies health-related quality of life[59]
- Patient-Reported Outcomes Measurement Information System (PROMIS). The PROMIS measures administered included:
  ○ *PROMIS® Item Bank v2.0 - Physical Function and PROMIS Item Bank v2.0 -Cognitive Function*, two computer adaptive surveys with tailored questionnaires based on item response theory[60].
  ○ *PROMIS Scale v1.2 - Global Health Mental 2a and* PROMIS Item Bank v1.0 - *Psychosocial Illness Impact−Positive* - Short Form 8a, two surveys with fixed questions[61–64].
  ○ *PROMIS Pool v1.0 - Dyspnea Time Extension* computer adaptive instrument for participants who reported shortness of breath[65–67]. This 7-item questionnaire assesses whether there has been a meaningful increase or decrease in the duration of time needed by an adult to perform a given task in the past 7 days compared to 3 months ago due to shortness of breath.

For all PROMIS measures, scoring was based on PROMIS standardized instructions and conversion to a t-statistic[68].
- Health Recovery Score: Overall health was also assessed by a health recovery score utilizing a Visual Analog Scale of 1–100 to indicate overall physical and mental function compared to pre-COVID function.

All data were reviewed centrally to ensure accuracy and consistency. Any data concerns were resolved by querying the site.

The full study data collection forms for the quarterly outpatient surveys are provided in the Supplementary Information (Surveys Administered).

## Assays

- SARS-CoV-2 viral load was assessed by a central laboratory from nasal swab samples at each defined time point by RT-PCR of the viral N1, and N2 genes[69] (Supplementary Methods).
- Anti-SARS-CoV-2 spike (S), and receptor binding domain (RBD) antibodies were quantitated by enzyme-linked immunosorbent assay (ELISA) in serum specimens[70] (Supplementary Methods).
- Autoantibodies: blocking antibodies against type I IFNs (IFNα, IFNβ, and IFNω) were assessed in a multiplex, particle-based assay (Supplementary Methods).
- Blood CyTOF 65 cell subsets were identified using a panel of 43 antibodies to cell surface markers expressed by distinct lineages and intracellular markers of functional status. A semi-automated gating strategy was used[71] (Supplementary Methods).
- Proximity Extension Assay (O-Link) multiplex assay inflammatory panel (Olink Bioscience, Uppsala, Sweden) includes 92 different proteins associated with human inflammatory conditions (Supplementary Methods).
- Plasma global metabolomics was assessed using liquid chromatography-mass spectrometry technique as described (Supplementary Methods).

## Statistics

### Convalescent clinical outcome assessment

**Overall analytic approach.** Because we focused on how longitudinal patterns in PROs might define clinical phenotypes relevant to PASC, we modeled each PRO using an approach that assumes the population is composed of distinct groups, each of which follows a different underlying and unobserved trajectory, with individual-level variation around that trajectory. Our approach to the high-dimensional data problem presented by multiple PROs captured across multiple timepoints was first to preserve longitudinal patterns within each PRO using LCMMs. We then reduced further the dimensionality of the data by clustering the resulting groupings across multiple PROs using five clustering algorithms and four diagnostic statistics. This approach mirrors a recent study[72] with similar aims to cluster high-dimensional data across several clinical outcomes. We compared the mean values of each PRO within each cluster to the remaining sample to interpret each cluster with respect to the contributions of each PRO. Finally, we compared clinical, demographic, and laboratory assay variables across each of the clusters thus determined. Convalescent clinical outcomes were analyzed using R Statistical Software version 4.2.1.

**Latent class analysis.** We considered the PROs collected at quarterly intervals and modeled longitudinally using LCMM, a family of models of which a well-known and commonly used example is the group-based trajectory model[73,74], implemented by R package "lcmm". Outcome variables were the EQ-5D-5L, global Health Recovery Score, PROMIS Cognitive Function Score, PROMIS Physical Function Score, PROMIS Dyspnea Score, PROMIS Global Mental Health Score, and PROMIS Psychosocial Illness Impact Positive Score. We evaluated linear and quadratic models with number of groupings ranging from 1 to 5, and specified the model based on convergence criteria and goodness-of-fit using Bayesian Information Criteria (BIC). For each outcome, we selected the model that converged and had the lowest BIC.

**Cluster analysis.** Using the assigned groups from the LCMM step for each PRO, we then applied cluster analysis to group participants with similar PRO longitudinal patterns. For those PROs with no distinct longitudinal clusters, we assigned to each participant the within-participant mean for that PRO. We calculated inter-participant similarity using Gower distance implemented by R package "CluMix". We applied five clustering algorithms (Ward, McQuitty, Average, PAM, and Complete) to the distance matrix to identify the optimal number of clusters, and selected the optimal model based on four cluster fitting statistics (within-cluster SS, average silhouette width, Dunn index, and ratio of within-to-between SS). We then excluded cluster solutions with degenerate clusters (e.g., those with only one participant).

Thus, the best model performed well on the four fitting statistics overall and had a clinically interpretable number of clusters. We further excluded solutions with clusters of size $n = 5$ or smaller. We generated cluster assignments using R package "cluster" with fit statistics implemented by package "fpc". To estimate the strength of association of each PRO with particular clusters, we calculated a t-statistic comparing the mean value of each PRO within each cluster versus the mean value of that PRO across the remaining clusters. The t-statistics were recoded such that negative values indicated a greater degree of patient-reported deficit, while positive values indicated no reported deficit.

**Statistical analysis—demographic & clinical variables.** We report median (interquartile range, IQR) for continuous variables and frequency (percent) for categorical variables. We examined bivariate associations between demographic and clinical factors and the PRO clusters using the Wilcoxon rank-sum test for continuous variables and chi-square test for categorical variables. Multinomial logistic regression was used to examine the adjusted associations between demographic and clinical factors and cluster membership, comparing the likelihood of being in each of the deficit clusters relative to MIN deficit cluster. $P < 0.05$ was considered statistically significant.

**Analysis of laboratory assays.** To identify modules of correlated features from high-dimensional 'omics data, we utilized Weighted Gene Co-expression Network Analysis (WGCNA) v1.71. We specified the module "value" as the first principal component of features in the module to summarize each group of assay readouts for subsequent analysis. For interpretation, the features in each module were annotated to biological processes by performing an enrichment analysis leveraging biological knowledge bases, including MSigDB Hallmark gene sets, SMPDB metabolites and pathways. To identify the associations between different immune measurements and the four PRO clusters, we used two complementary approaches that each account for repeated measures per individual. The first was the use of generalized linear mixed effects models (GLMs) including a random effect for individual but not accounting for the timing of sample collection, the second was generalized additive mixed effects models (GAMs) that do account for timing of sample collection and allowed us to investigate longitudinal patterns. For both approaches, we utilized the measurements from samples collected within 28 days of hospital admission (with up to 6 samples per participant). In the GLM approach, we ignored the time of sample collection and identified features with different mean values from the aggregated timepoints among the PRO clusters. In the GAM approach, we investigated whether there were either differences in the average values over time or differences in the temporal patterns of features among PRO clusters. Each model is adjusted for fixed effects of participant age, sex, and random effects for participant and enrollment site. We used R packages, "lme4" for the GLM approach and "gamm4" for GAM approach. Significant associations were defined at false discovery rate (FDR) < 0.05 using the Benjamini-Hochberg method to account for multiple comparisons. For both approaches, significant features were tested by post-hoc pairwise comparisons to identify the differences between each pair of PRO clusters to facilitate interpretation. Features for which the aggregated mean values in the GLM, the average over time (referred to as intercept

in the gamm4 documentation), or the shape (referred to as the smoothing term in the gamm4 documentation) differed among PRO clusters at FDR < 5% were considered significant.

**Case-control analysis: anti-IFN antibodies.** To determine whether IFN autoantibodies were associated with viral burden, we performed a case-control analysis, and identified age and sex-matched controls for the 24 individuals who tested positive for IFN autoantibodies with blocking activity at their earliest hospital visit (3:1 ratio of controls to cases). SARS-CoV-2 viral load (N1 Ct and N2 Ct values), SARS-CoV-2 RBD, and Spike binding IgG titers were compared between cases and controls. Significant differences in median levels between the two groups were assessed using Wilcoxon rank-sum test.

### Reporting summary
Further information on research design is available in the Nature Portfolio Reporting Summary linked to this article.

## Data availability
The IMPACC Data Sharing Plan is designed to enable the widest dissemination of data, while also protecting the privacy of the participants and the utility of the data by de-identifying and masking potentially sensitive data elements. All IMPACC data, including those generated in this study, have been deposited in the Immunology Database and Analysis Portal (ImmPort), a NIAID Division of Allergy, Immunology and Transplantation funded data repository under accession code SDY1760. All raw and processed data are available under restricted access to comply with the NIH public data sharing policy for IRB-exempted public health surveillance studies, access can be obtained via AccessClinicalData@NIAID (https://accessclinicaldata.niaid.nih.gov/study-viewer/clinical_trials). Additional guidelines for access are outlined on ImmPort (https://docs.immport.org/home/impaccslides). In addition, raw LC-MS data for metabolomics are submitted to Metabolights repository under accession number MTBLS850.

## Code availability
All codes for the analyses and tables generated by this study have been deposited in the Bitbucket repository https://bitbucket.org/kleinstein/impacc-public-code/src/master/convalescent_manuscript/ and are publicly available as of the date of publication.

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

## Acknowledgements

The study was funded by supplements to project grants and cooperative agreements awarded by the National Institute of Allergy and Infectious Disease (NIAID), including 5R01AI135803-03 (D.B.C., F.K.), 5U19AI118608-04 (O.L., H.S., L.R.B., J.D.A.), 5U19AI128910-04 (E.H., C.B.C., R.P.S., G.A.M.), 4U19AI090023-11 (B.P., N.R., S.B.), 4U19AI118610-06 (V.S., A.F.S., F.K., H.V.B., S.K.S.), R01AI145835-01A1S1 (W.B.M., C.L.H.), 5U19AI062629-17 (J.P.M., N.I.A.H.), 5U19AI057229-17 (M.M.D., K.C.N., H.M.), 5U19AI125357-05 (M.K., C.B.), 5U19AI128913-03 (J.S., F.R.), 3U19AI077439-13 (D.E., C.C., C.R.L., W.E.), 5U54AI142766-03 (M.A.A., S.C.B.), 5R01AI104870-07 (L.E., E.M.), 3U19AI089992-09 (D.H., R.R.M., A.C.S.), and 5R01AI132774-03 (M.C.A.). NIAID project scientists participated collaboratively in study design, data analyses, interpretation, and writing of the report. The manuscript content is solely the responsibility of the authors and does not necessarily represent the official views of the NIH.

## Author contributions

A.O., P.B., M.C.A., N.R., A.A., L.B., C.S.C., G.M., C.B.C., V.S., C.H., K.C.N., J.S., A.S., D.C., N.I.A.H., C.B., S.C.B., E.M., and V.S.M. were involved in the study concept and design. A.O., C.M., N.D.J., H.V.B., M.C.A., D.E., S.L., F.K.R., J.D.-A., P.B., and N.R. contributed to the study methodology. C.M., N.D.J., H.V.B., S.L., F.K.R., L.B.R., S.K.S., C.R.L., W.E., S.E.B., H.T.M. were involved in data curation. L.B., C.S.C., G.M., C.B.C., N.R., V.S., C.H., K.C.N., J.M., A.S., D.C., N.I.A.H., C.B., S.C.B., and E.M. contributed to the investigation. C.M., N.D.J., H.VB., S.L., A.O., J.-D.A., J.M., and V.S.M. provided software. A.O., C.M., N.D.J., H.V.B., M.C.A., D.E., S.L., F.K.R., J.Q., C.S., R.D., and C.P.S. were involved in formal analysis. A.O., C.M., N.D.J., H.V.B., M.C.A., D.E., S.L., and F.K.R. were involved in data validation, visualization, or replication of analyses. A.O., C.M., M.C.A., J.S., N.R., and P.B. drafted the manuscript. A.O., J.S., N.D.J., C.M., C.S., C.B.C., M.K., L.B., A.S., F.K.R., H.V.B., D.E., S.L., A.F.S., V.S., D.H., R.M., S.K., O.L., C.B., E.H., D.J.E., B.P.U., K.C.N., M.D., C.H., W.M., N.I.A.H., J.M., M.A.A., S.C.B., D.C., F.KH., L.I.E., E.M., G.M., R.S., J.D.-A., B.P.E., A.A., E.R., M.C.A., P.B., H.S., L.N.G., N.R., and C.P.S. critically reviewed and revised the manuscript. P.B., A.A., A.O., S.K., B.P.E., N.R., E.R., J.-D-A., and J.M. were involved in project administration, and A.O., J.S., N.D.J., C.M., C.S.C., C.B.C., M.K., L.B., A.S., F.K.R., H.V.B., D.E., S.L., A.F.S., V.S., D.H., R.M., S.K., O.L., C.B., E.H., D.E., B.P.U., K.C.N., M.D., C.H., W.M., N.I.A.H., J.M., M.A.A., S.C.B., D.C., F.K.H., L.I.E., E.M., G.M., R.S., J.D.-A., B.P.E., A.A., E.R., M.C.A., P.B., and N.R. provided study resources. O.L., R.S., E.H., A.F.S., W.M., M.D., B.P.U., E.F., D.J.E., R.M., D.H., F.K.H., J.M., M.K., M.A.A., and L.I.E. obtained funding. P.B., A.O., M.C.A., and N.R. provided supervision.

## Competing interests

M.C.A. has received grant support from NIH-R01AI32774 for this project funded through this R01 and travel fees from NIAID for travel to the American Thoracic Society 2022 to present data related to this study. M.A.A. has received funding from NIH, NIAID-5U54AI142766-03 through institution. L.R.B. has received grant support from NIH NIAID, through institution. P.M.B. and A.D.A. federal employees serving as a project scientist for this project but had no role in funding decisions or oversight of relevant grants. B.P. has received funding from NIH/NIAID. C.B.C. has received funding from NIAID with payments to institution (Drexel University), Bill & Melinda Gates Foundation for COVID-19 work paid to institution, consulting fees from bioMerieux on clinical biomarkers, serves as DSMB, Advisory board for Convalescent Plasma COVID-19 study for the National Heart, Lung and Blood Institute (NHLBI), and is acting Leadership as President Board of Directors for the National Foundation of Emergency Medicine (NFEM), a non-profit supporting emergency medicine research and researchers. C.C. has received funding from NHLBI, grants from Bayer, Roche-Genentech, Quantum Leap Healthcare Collaborative, and consulting fees from Vasomune, Gen1e Life Sciences, Cellenkos, Janssen. L.E. has received grant funding from NIH R01AI104870-S1. D.E. has received NIH grants awarded to institution (UCSF). C.L.H. has received funding from NIH and the American Lung Association, travel support from Stanford, Harvard, Critical Care Clinical Trialists, Critical Care Reviews, and the University of Michigan, serves as DSMB Quantum Health for iSPY COVID, and paid participation as member, Board of Directors for American Thoracic Society. J.P.M. has received funding from NIH Grant # 3U19AI0626629-17S2. S.H.K. has received paid consulting fees from Peraton for personal consulting related to Immport data repository. M.K. has received grants support from NIAID through institution (University of Arizona). F.K.R. has received funding from NIAID Collaborative Influenza Vaccine Innovation Centers (CIVIC) contract 75N93019C00051, JPB Foundation and the Open Philanthropy Project (research grant 2020-215611, 5384), National Cancer Institute, NIH contract no.75N91019D00024, Task order no. 75N91020F00003, research funding from Pfizer for development of animal models for SARS-CoV-2, consulting fees from Pfizer, Seqirus, Avimex, Third Rock Ventures, paid lecture fees, and patents filed at the Icahn School of Medicine at Mount Sinai relating to SARS-CoV-2 serological assays (the "Serology Assays" and NDV-based SARS-CoV-2 vaccines which list F.K.R. as co-inventor). G.A.M. has received consulting fees from Gilead. E.M. has received funding from the NIH IMPACC R01AI104870-S1, grants from Babson Diagnostics, K0826-1616-11 through Institution Dell Medical School at UT Austin, paid speaker fees from MS Association of America, and serves as DSMB for Advisory boards of Genentech, Horizon, Teva and Viela Bio. W.B.M. has received funding from NIH NIAID R01AI14583. R.R.M. has contracts and grants for IMPACC study from NIAID AI 089992 aid and a Leadership Councilor role 2018-2021 for Society of Leukocyte Biology. K.C.N. has research funding from the National Heart, Lung, and Blood Institute (NHLBI), National Institute of Environmental Health Sciences (NIEHS), Food Allergy Research & Education (FARE), Director of the World Allergy Organization center of Excellence for Stanford and National Institute of Allergy and Infectious Diseases (NIAID) through institution, paid participation for service on Data Safety Monitoring Board of Director from World Allergy Organization Center of Excellence for Stanford, earns stocks as co-founder at Seed Health, IgGenix, ClostraBio, ImmuneID and financial interests as advisor and co-founder for Cour Pharma, Before Brands, Alladapt, and Latitude, is a national scientific committee member for Network ITN, NIH clinical research centers, and has listed Patents: Mixed allergen composition and methods for using the same Granulocyte-based methods for detecting and monitoring immune system disorders, licensee: (Alladapt and Before Brands no: US15/048,609); Application number: US12/610,94 Methods and Assays for Detecting and Quantifying Pure Subpopulations of White blood cells in immune system disorders. O.L. has received NIH/NIAID grants through institution for Human Immunology Project Consortium Funding (U19) 1-U19-AI118608-01A1 as PI Role and support as a speaker for presentation regarding the Coronavirus pandemic from Midsized Bank Coalition of Americ (MBCA) and Moody's Analytics. E.F.R. has received grants supported by NIAID U19AI12891303. N.G.R. has research grants from NIH, Pfizer, Merck, Sanofi, Quidel, and Lilly, and serves on safety committees for ICON and EMMES and the advisory boards of Moderna and Sanofi. Her institution has also received funding from NIH to conduct clinical trials of COVID-19

vaccines. J.S. received a grant from NIAID U19 for study implementation. A.C.S. has financial support from NIH U19 AI089992, NIH K24 AG042489. V.S. filed Patents at the Icahn School of Medicine at Mount Sinai relating to SARS-CoV-2 serological assays (the "Serology Assays" and NDV-based SARS-CoV-2 vaccines, which list V.S. as co-inventor. H.B. has received funding from NIH (Dengue Human Immunology Project Consortium - Mount Sinai IMPACC COVID-19 Cores), U19 AI118610 S1, NIH, CEIRR,75N93021C00014. SEB All funding sources are from NIH U19 AI090023-1S1, paid honoraria for serving on SAB for NIH NIAID P01AI174856-01, travel support from the University of Manitoba, and received TAK-242 from TAKEDA for pre-clinical testing. H.S. has received funding from 3 U19 AI 118608-05S3 through institution and travel support from NIAID. L.G. has received funding from Sean N. Parker Center for Allergy & Asthma Research, grants from Pfizer, and consulting fees from UnitedHealth Group. H.M. has received funding from NIH grant 2U19AI057229. R.D. has received grant support from UCSF COVID-19 Immunophenotyping Clinical Study and Core Laboratories (Grant number U19AI077439). V.S.M. has grants paid consulting as Vice Chair for EveryLife Foundation, Advisory board for Gemini labs, and stocks at My Own Med, Inc., d/b/a Respond Health. The remaining authors declare no competing interests.

## Additional information

Al Ozonoff [1,22], Naresh Doni Jayavelu[2,22], Shanshan Liu[1,22], Esther Melamed[3,22], Carly E. Milliren[1,22], Jingjing Qi[4,22], Linda N. Geng[5,22], Grace A. McComsey[6], Charles B. Cairns [7], Lindsey R. Baden[8], Joanna Schaenman[9], Albert C. Shaw[10], Hady Samaha [11], Vicki Seyfert-Margolis[12], Florian Krammer [4], Lindsey B. Rosen[13], Hanno Steen [8], Caitlin Syphurs[1], Ravi Dandekar[14], Casey P. Shannon [15], Rafick P. Sekaly[6], Lauren I. R. Ehrlich [3], David B. Corry[16], Farrah Kheradmand [16], Mark A. Atkinson[17], Scott C. Brakenridge[17], Nelson I. Agudelo Higuita[18], Jordan P. Metcalf[18], Catherine L. Hough[19], William B. Messer [19], Bali Pulendran [5], Kari C. Nadeau [5], Mark M. Davis [5], Ana Fernandez Sesma [4], Viviana Simon [4], Harm van Bakel [4], Seunghee Kim-Schulze[4], David A. Hafler [10], Ofer Levy [8], Monica Kraft[20], Chris Bime[20], Elias K. Haddad[7], Carolyn S. Calfee[14], David J. Erle[14], Charles R. Langelier[14], Walter Eckalbar[14], Steven E. Bosinger [11], IMPACC Network*, Bjoern Peters [21], Steven H. Kleinstein [10], Elaine F. Reed[9], Alison D. Augustine[13], Joann Diray-Arce[1], Holden T. Maecker [5], Matthew C. Altman [2,23], Ruth R. Montgomery [10,23], Patrice M. Becker [13,23] & Nadine Rouphael [11,23] ✉

[1]Clinical & Data Coordinating Center (CDCC), Precision Vaccines Program, Boston Children's Hospital, Boston, MA, USA. [2]Benaroya Research Institute, Seattle, WA, USA. [3]The University of Texas at Austin, Austin, TX, USA. [4]Icahn School of Medicine at Mount Sinai, New York, NY, USA. [5]Stanford University, Stanford, CA, USA. [6]Case Western Reserve University and University Hospitals of Cleveland, Cleveland, OH, USA. [7]Drexel University/Tower Health Hospital, Philadelphia, PA, USA. [8]Boston Clinical Site: Precision Vaccines Program, Boston Children's Hospital, Brigham and Women's Hospital, and Harvard Medical School, Boston, MA, USA. [9]David Geffen School of Medicine at the University of California Los Angeles, Los Angeles, CA, USA. [10]Yale School of Medicine, and Yale School of Public Health, New Haven, CT, USA. [11]Emory University, Atlanta, GA, USA. [12]MyOwnMed, Inc, Bethesda, MD, USA. [13]National Institute of Allergy and Infectious Diseases/National Institutes of Health, Bethesda, MD, USA. [14]University of California San Francisco School of Medicine, San Francisco, CA, USA. [15]Centre for Heart Lung Innovation, Providence Research, St. Paul's Hospital, and the PROOF Centre of Excellence, Vancouver, BC, Canada. [16]Baylor College of Medicine, and the Center for Translational Research on Inflammatory Diseases, Michael E. DeBakey VA Medical Center, Houston, TX, USA. [17]University of Florida/University of South Florida, Tampa, FL, USA. [18]Oklahoma University Health Sciences Center, Oklahoma City, OK, USA. [19]Oregon Health & Science University, Portland, OR, USA. [20]University of Arizona, Tucson, AZ, USA. [21]La Jolla Institute for Immunology, La Jolla, CA, USA. [22]These authors contributed equally: Al Ozonoff, Naresh Doni Jayavelu, Shanshan Liu, Esther Melamed, Carly E. Milliren, Jingjing Qi, Linda N. Geng. [23]These authors jointly supervised this work: Matthew C. Altman, Ruth R. Montgomery, Patrice M. Becker, Nadine Rouphael. ✉e-mail: nroupha@emory.edu

## IMPACC Network

**IMPACC Steering Committee** Al Ozonoff [1,22], Joann Diray-Arce[1], Matthew C. Altman [2,23], Lauren I. R. Ehrlich [3], Esther Melamed[3,22], Ana Fernandez Sesma [4], Viviana Simon [4], Bali Pulendran [5], Kari C. Nadeau [5], Mark M. Davis [5], Grace A. McComsey[6], Rafick P. Sekaly [6], Charles B. Cairns [7], Elias K. Haddad [7], Lindsey R. Baden[8], Ofer Levy [8], Joanna Schaenman[9], Elaine F. Reed[9], Albert C. Shaw[10], David A. Hafler [10], Ruth R. Montgomery [10,23], Steven H. Kleinstein [10], Nadine Rouphael [11,23]✉, Patrice M. Becker [13,23], Alison D. Augustine[13], Carolyn S. Calfee[14], David J. Erle [14], David B. Corry[16], Farrah Kheradmand [16], Mark A. Atkinson[17], Scott C. Brakenridge[17], Nelson I. Agudelo Higuita[18], Jordan P. Metcalf[18], Catherine L. Hough[19], William B. Messer [19], Monica Kraft[20], Chris Bime[20] & Bjoern Peters [21]

**Clinical & Data Coordinating Center (CDCC)** Al Ozonoff [1,22], Carly E. Milliren[1,22], Joann Diray-Arce[1], Caitlin Syphurs[1], Kerry McEnaney[1], Brenda Barton[1], Claudia Lentucci[1], Mehmet Saluvan[1], Ana C. Chang[1], Annmarie Hoch[1], Marisa Albert[1], Tanzia Shaheen[1], Alvin T. Kho[1], Shanshan Liu[1,22], Sanya Thomas[1], Jing Chen[1], Maimouna D. Murphy[1], Mitchell Cooney[1], Arash Nemati Hayati[1], Robert Bryant[1] & James Abraham[1]

**IMPACC Data Analysis Group** Al Ozonoff [1,22], Joann Diray-Arce[1], Naresh Doni Jayavelu[2,22], Matthew C. Altman [2,23], Scott Presnell[2], Tomasz Jancsyk[2], Cole Maguire[3], Jingjing Qi[4,22], Brian Lee[4], Slim Fourati[6], Charles B. Cairns [7], Denise A. Esserman[10], Leying Guan[10], Steven H. Kleinstein[10], Jeremy Gygi[10], Shrikant Pawar[10], Anderson Brito[10], Gabriela K. Fragiadakis[14], Ravi Patel[14], Casey P. Shannon [15], Scott J. Tebbutt[15], Bjoern Peters [21], James A. Overton[21], Randi Vita[21] & Kerstin Westendorf[21]

**IMPACC Site Investigators** Lauren I. R. Ehrlich [3], Esther Melamed[3,22], Rama V. Thyagarajan[3], Justin F. Rousseau[3], Dennis Wylie[3], Todd A. Triplett[3], Erna Kojic[4], Viviana Simon [4], Kari C. Nadeau [5], Sharon Chinthrajah[5], Neera Ahuja[5], Angela J. Rogers[5], Maja Artandi[5], Linda N. Geng[5,22], Grace A. McComsey[6], George Yendewa[6], Charles B. Cairns [7], Debra L. Powell[7], James N. Kim[7], Brent Simmons[7], I. Michael Goonewardene[7], Cecilia M. Smith[7], Mark Martens[7], Lindsey R. Baden[8], Amy C. Sherman[8], Stephen R. Walsh[8], Nicolas C. Issa[8], Joanna Schaenman[9], Ramin Salehi-Rad[9], Albert C. Shaw[10], Charles Dela Cruz[10], Shelli Farhadian[10], Akiko Iwasaki[10], Albert I. Ko[10], Nadine Rouphael [11,23]✉, Evan J. Anderson[11], Aneesh K. Mehta[11], Jonathan E. Sevransky[11], Vicki Seyfert-Margolis[12], Carolyn S. Calfee[14], David J. Erle [14], Aleksandra Leligdowicz[14], Michael A. Matthay[14], Jonathan P. Singer[14], Kirsten N. Kangelaris[14], Carolyn M. Hendrickson[14], Matthew F. Krummel[14], Charles R. Langelier [14], Prescott G. Woodruff[14], David B. Corry[16], Farrah Kheradmand [16], Scott C. Brakenridge[17], Matthew L. Anderson[17], Faheem W. Guirgis[17], Nelson I. Agudelo Higuita[18], Jordan P. Metcalf[18], Douglas A. Drevets[18], Brent R. Brown[18], William B. Messer [19], Sarah A. R. Siegel[19], Zhengchun Lu[19], Monica Kraft[20], Chris Bime[20], Jarrod Mosier[20] & Hiroki Kimura[20]

**IMPACC Core Laboratory** Joann Diray-Arce[1], Matthew C. Altman [2,23], Bernard Khor[2], Florian Krammer [4], Harm van Bakel [4], Adeeb Rahman[4], Daniel Stadlbauer[4], Jayeeta Dutta[4], Seunghee Kim-Schulze[4], Ana Silvia Gonzalez-Reiche[4], Adriana van de Guchte[4], Juan Manuel Carreño[4], Gagandeep Singh[4], Ariel Raskin[4], Johnstone Tcheou[4], Dominika Bielak[4], Hisaaki Kawabata[4], Brian Lee[4], Hui Xie[4], Geoffrey Kelly[4], Manishkumar Patel[4], Kai Nie[4], Temima Yellin[4], Miriam Fried[4], Leeba Sullivan[4], Sara Morris[4], Holden T. Maecker [5], Scott Sieg[6], Hanno Steen [8], Patrick van Zalm[8], Benoit Fatou[8], Kevin Mendez[8], Jessica Lasky-Su[8], Scott R. Hutton[8], Greg Michelotti[8], Kari Wong[8], Meenakshi Jha[8], Arthur Viode[8], Naama Kanarek[8], Boryana Petrova[8], Albert C. Shaw[10], Yujiao Zhao[10], Charles Dela Cruz[10], Ruth R. Montgomery [10,23], Steven E. Bosinger [11], Arun K. Boddapati[11], Greg K. Tharp[11], Kathryn L. Pellegrini[11], Elizabeth Beagle[11], David Cowan[11], Sydney Hamilton[11], Susan Pereira Ribeiro[11], Thomas Hodder[11], Lindsey B. Rosen[13], Serena Lee[13], Charles R. Langelier [14], Michael R. Wilson[14], Ravi Dandekar[14], Bonny Alvarenga[14], Jayant Rajan[14], Walter Eckalbar[14], Andrew W. Schroeder[14], Alexandra Tsitsiklis[14], Eran Mick[14], Yanedth Sanchez Guerrero[14], Christina Love[14], Lenka Maliskova[14] & Michael Adkisson[14]

**IMPACC Clinical Study Team** Cole Maguire[3], Nadia Siles[3], Janelle Geltman[4], Kerin Hurley[3], Miti Saksena[4], Deena Altman[4], Erna Kojic[4], Komal Srivastava[4], Lily Q. Eaker[4], Maria C. Bermúdez-González[4], Katherine F. Beach[4], Levy A. Sominsky[4], Arman R. Azad[4], Lubbertus C. F. Mulder[4], Giulio Kleiner[4], Alexandra S. Lee[5], Evan Do[5], Andrea Fernandes[5], Monali Manohar[5], Thomas Hagan[5], Catherine A. Blish[5], Hena Naz Din[5], Jonasel Roque[5], Samuel Yang[5], Natalia Sigal[5], Iris Chang[5], Heather Tribout[6], Paul Harris[6], Mary Consolo[6], Jennifer Connors[7], Mariana Bernui[7], Michele A. Kutzler[7], Carolyn Edwards[7], Edward Lee[7], Edward Lin[7], Brett Croen[7], Nicholas C. Semenza[7], Brandon Rogowski[7], Nataliya Melnyk[7], Kyra Woloszczuk[7], Gina Cusimano[7], Mathew R. Bell[7], Sara Furukawa[7], Renee McLin[7], Pamela Schearer[7], Julie Sheidy[7], George P. Tegos[7],

Crystal Nagle[7], Ofer Levy [8], Kinga Smolen[8], Michael Desjardins[8], Simon van Haren[8], Xhoi Mitre[8], Jessica Cauley[8], Xiaofang Li[8], Alexandra Tong[8], Bethany Evans[8], Christina Montesano[8], Jose Humberto Licona[8], Jonathan Krauss[8], Jun Bai Park Chang[8], Natalie Izaguirre[8], Rebecca Rooks[8], David Elashoff[9], Jenny Brook[9], Estefania Ramires-Sanchez[9], Megan Llamas[9], Adreanne Rivera[9], Claudia Perdomo[9], Dawn C. Ward[9], Clara E. Magyar[9], Jennifer A. Fulcher[9], Harry C. Pickering[9], Subha Sen[9], Omkar Chaudhary[10], Andreas Coppi[10], John Fournier[10], Subhasis Mohanty[10], M. Catherine Muenker[10], Allison Nelson[10], Khadir Raddassi[10], Michael Rainone[10], William E. Ruff[10], Syim Salahuddin[10], Wade L. Schulz[10], Pavithra Vijayakumar[10], Haowei Wang[10], Elsio Wunder Jr.[10], H. Patrick Young[10], Yujiao Zhao[10], Jessica Rothman[10], Anna Konstorum[10], Ernie Chen[10], Chris Cotsapas[10], Nathan D. Grubaugh[10], Xiaomei Wang[10], Leqi Xu[10], Hiromitsu Asashima[10], Laurel Bristow[11], Laila Hussaini[11], Kieffer Hellmeister[11], Hady Samaha [11], Sonia Tandon Wimalasena[11], Andrew Cheng[11], Christine Spainhour[11], Erin M. Scherer[11], Brandi Johnson[11], Amer Bechnak[11], Caroline R. Ciric[11], Lauren Hewitt[11], Erin Carter[11], Nina Mcnair[11], Bernadine Panganiban[11], Christopher Huerta[11], Jacob Usher[11], Vicki Seyfert-Margolis[12], Tatyana Vaysman[13], Steven M. Holland[13], Yumiko Abe-Jones[14], Saurabh Asthana[14], Alexander Beagle[14], Sharvari Bhide[14], Sidney A. Carrillo[14], Suzanna Chak[14], Gabriela K. Fragiadakis[14], Rajani Ghale[14], Ana Gonzalez[14], Alejandra Jauregui[14], Norman Jones[14], Tasha Lea[14], Deanna Lee[14], Raphael Lota[14], Jeff Milush[14], Viet Nguyen[14], Logan Pierce[14], Priya A. Prasad[14], Arjun Rao[14], Bushra Samad[14], Cole Shaw[14], Austin Sigman[14], Pratik Sinha[14], Alyssa Ward[14], Andrew Willmore[14], Jenny Zhan[14], Sadeed Rashid[14], Nicklaus Rodriguez[14], Kevin Tang[14], Luz Torres Altamirano[14], Legna Betancourt[14], Cindy Curiel[14], Nicole Sutter[14], Maria Tercero Paz[14], Gayelan Tietje-Ulrich[14], Carolyn Leroux[14], Ravi Patel[14], Neeta Thakur[14], Joshua J. Vasquez[14], Lekshmi Santhosh[14], Li-Zhen Song[16], Ebony Nelson[16], Lyle L. Moldawer[17], Brittany Borresen[17], Brittney Roth-Manning[17], Ricardo F. Ungaro[17], Jordan Oberhaus[17], J. Leland Booth[18], Lauren A. Sinko[18], Amanda Brunton[19], Peter E. Sullivan[19], Matthew Strnad[19], Zoe L. Lyski[19], Felicity J. Coulter[19], Courtney Micheleti[19], Michelle Conway[20], Dave Francisco[20], Allyson Molzahn[20], Heidi Erickson[20], Connie Cathleen Wilson[20], Ron Schunk[20], Bianca Sierra[20] & Trina Hughes[20]

