## [Peer Review File · Nature Communications]

Features of acute COVID-19 associated with Post-Acute Sequelae of SARS-CoV-2 phenotypes: results from the IMPACC studyReviewers' Comments:

Reviewer #1:

Remarks to the Author:

This work studies the associations between acute COVID-19 phenotypes with post-acute sequelae of the disease (PASC). A cohort of 590 COVID-19 inpatients (from early waves of the pandemic) were followed up for one year after discharge. For identifying subtypes of PASCs, clustering analyses were conducted on patient reported outcomes. Associations of demographic/clinical characteristics and acute COVID-19 phenotypes in the hospitalisation were then studied within/between these PASC subtypes.

The strength of the study include: good follow-up study design with a descent period, comprehensive data collection and processing, and reasonable technical work including clustering methods and statistical analysis. The clinical findings are overall inline with those from other investigations and complement existing ones with detailed analysis results on acute phenotypes' associations of four subtypes of PASCs.

The main limitation of this work resides in the study period which was in the early time of the COVID-19 pandemic. This makes the clinical findings not directly actionable/informative to the latest development of the virus/variants, the change of treatments/managements/public health policies and the wide rollout vaccinations. Also, as the authors pointed out by themselves, external validation study would have made the findings more robust.

Some specific comments:

1. It would be great to report detailed results of the latent analysis results on patient reported outcomes.
2. The Ward algorithm identified six clusters (p. 459). It would be great to see the details of these clusters. Given that only four clusters (including one MIN cluster) were included in later analysis, apparently, two clusters were merged. It is important to see some details on these clusters and the t-tests on these for concluding that they were with minimal PASC deficit.
3. It is not clear what the numbers in Figure 3 and Figure 4S denote.
4. in line 482, "(FIGURE 4S, ...)" => should this be FIGURE 3 because the whole paragraph starts with discussing "N1 gene SARS-CoV-2 PCR cycle threshold (Ct)" and N1 Ct is presented in FIGURE 3 while N2 Ct is in the supplementary?
5. It would be great to also see p-values in the t-test analysis of the PRO clusters (i.e., Table 1s).
6. In the conclusion, line 544, the authors claim "we found no association between PASC and acute COVID-19 disease severity". It would be nice to explicitly point out which experiments conclude this in this study? The same applies to lines 563-564 "use of remdesivir and steroids in the inpatient period was not associated with a decrease in PASC prevalence."

Reviewer #2:

Remarks to the Author:

This manuscript describes results from a major prospective study on post-acute sequelae for COVID-19 (PASC). PASC research is important for public health, and contributes to our more general understanding of post-viral sequelae. Study strengths include multiple sites, inclusion of deep phenotype data from acute COVID-19, and longitudinal patient-reported outcomes.

Major comments

The latent class analysis and cluster analysis approach is an interesting and reasonable approach for selecting outcomes for PASC, a condition that still lacks a clear definition and mechanistic subtyping.

In the methods section for latent class analysis, there should be a description of how the assumptions of indicator variable independence was tested, and if needed, addressed.

The authors note that "In addition, we did not attempt to identify alternative causes of persistent or new symptoms. However, PRO measures were chosen to attempt to mitigate this limitation by including comparison to pre-illness baseline or some other appropriate recall period when possible." In a cohort, such as the one described, you would expect new onset symptoms over a year, even without COVID-19. It would be valuable to 1) follow up with review of medical records or survey questions to inquire about alternative diagnosis that might explain PROs and/or at least 2) note incidence of new onset or worsening PROs from literature on post-hospitalization.

Supplement page 3 link for code goes to a page that says "Repository is not available"
https://bitbucket.org/kleinstein/impacc-public-code/src/master/clinical_manuscript/

Minor comments

In Methods - Statistics, and in the legend for Table 1, the authors state that t-statistics were "recoded such that positive values indicate a greater degree of patient-reported deficit". However, the minimal deficit (MIN) group has the highest positive values. Should this say "recoded such that negative values..."?

Under Methods, Data Collection, Study Variables, and Biologic Samples, the complications do not appear to be defined in the previous IMPACC paper or the IMPACC protocol. In particular, given the results highlighted, it would be valuable to know whether patients had secondary bacterial infections, sepsis and/or kidney injury.

The study would be strengthened with information about whether patients had been on medications that affect the frequency of circulating B lymphocytes (rituximab, or other anti-CD20 agents).

The study would be strengthened by comparison to controls: 1) controls who did not have COVID, 2) patients hospitalized for a non-COVID-19 respiratory viral infection and/or 3) patients hospitalized for elective procedures where the length of stay is similar to PASC. While prospective longitudinal studies may be outside the scope of the current study, it would be helpful to note the limitations of not having these controls when drawing conclusions. For example, conclusions about methylhistidine pathways and long-term patient centered outcomes might reflect something specific to PASC, or factors common to many hospitalizations (stress metabolism, change in muscle tissue, decreased renal function).

For supplemental figures 4S-9S it would be helpful to use asterisks * or some similar convention to highlight which results are significant.

As a side note, degenerate clusters might be particularly interesting as outlier cases for study.

Reviewer #3:

Remarks to the Author:

- Thank you for the opportunity to review this paper! The authors describe a focused study in which

they created endotypes of long-Covid and assessed whether certain demographics, clinical characteristics, and viral assays were associated with cluster membership. A number of robust statistical methods were employed to handle the high-dimensional data. The methods & results are well-described. The paper is easy to read. I find no significant methodological limitations or flaws. I applaud the authors on focusing so much on patient-reported outcomes - there is a significant need for more research in this area. I have some minor comments/questions below:

- Study Participants - can you add the dates of the study to provide more context of when in the pandemic the study occurred?
- The methods seem reasonable, and it's clear a lot of thought went into the best way to handle the complicated dimensionality. Did you consider using a group-based trajectory model for grouping the longitudinal data, and if so, can you speak to why you didn't use that approach instead?
- Results - I think you're missing a number regarding the age & BMI where you have written "57 years (IQR 19)," & "31.8 kg/m² (IQR 9.6)", respectively - what's the range? It looks like this was done for length of stay, too - please list the 25th & 75th quantiles when reporting the IQR.
- Can you provide examples of what counts as an inpatient "complication"?
- In your statistical analysis plan, you noted using a multinomial logistic regression for the demographic clinical characteristics associated with each group; however, in Figure 2, you called this a proportional odds model. Can you clarify which regression method you used?
- In Figure 3, you have a black horizontal line with a number between several clusters - can you state what this number/line represents?

Thank you for allowing us the opportunity to revise the initial submission of our paper entitled "Features of acute COVID-19 associated with Post-Acute Sequelae of SARS-CoV-2 phenotypes: results from the IMPACC study". Below in *red italics* is our point-by-point responses to the reviewers comments.

Reviewer #1:

This work studies the associations between acute COVID-19 phenotypes with post-acute sequelae of the disease (PASC). A cohort of 590 COVID-19 inpatients (from early waves of the pandemic) were followed up for one year after discharge. For identifying subtypes of PASCs, clustering analyses were conducted on patient reported outcomes. Associations of demographic/clinical characteristics and acute COVID-19 phenotypes in the hospitalization were then studied within/between these PASC subtypes.

The strength of the study include: good follow-up study design with a descent period, comprehensive data collection and processing, and reasonable technical work including clustering methods and statistical analysis. The clinical findings are overall in line with those from other investigations and complement existing ones with detailed analysis results on acute phenotypes' associations of four subtypes of PASCs.

The main limitation of this work resides in the study period which was in the early time of the COVID-19 pandemic. This makes the clinical findings not directly actionable/informative to the latest development of the virus/variants, the change of treatments/managements/public health policies and the wide rollout vaccinations. Also, as the authors pointed out by themselves, external validation study would have made the findings more robust.

We appreciate all comments provided.

Some specific comments:

1. It would be great to report detailed results of the latent analysis results on patient reported outcomes.

Plots of the trajectory groups from LCMMs for each PRO are provided in Supplementary Figure 3S panels A-G. Detailed cluster fitting statistics are shown in Figure 3S panel H. We have added additional information on the group frequencies for each PRO in the appropriate Figure 3S panel caption.

2. The Ward algorithm identified six clusters (p. 459). It would be great to see the details of these clusters. Given that only four clusters (including one MIN cluster) were included in later analysis, apparently, two clusters were merged. It is important to see some details on these clusters and the t-tests on these for concluding that they were with minimal PASC deficit.

We have added a table (Table 1A-S) comparing the original six clusters, where we note few significant differences between the clusters collapsed into the minimal deficit cluster.

3. It is not clear what the numbers in Figure 3 and Figure 4S denote.

For the N1 (or N2) Ct the y axis numbers reflect the N1 or N2 gene SARS-CoV-2 PCR cycle threshold values and for the anti-RBG IgG (or anti-spike IgG) these numbers are the area under the curve. This information will be added to the legend of both figures.

4. in line 482, "(FIGURE 4S, ...)" => should this be FIGURE 3 because the whole paragraph starts with discussing "N1 gene SARS-CoV-2 PCR cycle threshold (Ct)" and N1 Ct is presented in FIGURE 3 while N2 Ct is in the supplementary?

N1 Ct data are presented in both Figure 3 and Figure 4S and therefore mentioned in the text accordingly.

5. It would be great to also see p-values in the t-test analysis of the PRO clusters (i.e., Table 1s).

We have added pairwise comparisons of PROs between clusters and across outcomes in Table 1S (now labeled Table 1B-S).

6. In the conclusion, line 544, the authors claim "we found no association between PASC and acute COVID-19 disease severity". It would be nice to explicitly point out which experiments conclude this in this study? The same applies to lines 563-564 "use of remdesivir and steroids in the inpatient period was not associated with a decrease in PASC prevalence."

"We found no association between PASC and acute COVID-19 disease severity" is based on the bivariate associations between PRO clusters and baseline SOFA score, baseline level of respiratory support, ICU stay, presence and number of complications (Table 1). We further added an evaluation of association between PRO cluster and disease severity, represented by the trajectory group (TGs) defined in previous work on the acute phase of illness. Finally, "use of remdesivir and steroids in the inpatient period was not associated with a decrease in PASC prevalence" is based on the same analysis as shown in Table 1. These were added to the results section.

Reviewer #2

This manuscript describes results from a major prospective study on post-acute sequelae for COVID-19 (PASC). PASC research is important for public health, and contributes to our more general understanding of post-viral sequelae. Study strengths include multiple sites, inclusion of deep phenotype data from acute COVID-19, and longitudinal patient-reported outcomes.

We thank the reviewer for the comments

Major comments

The latent class analysis and cluster analysis approach is an interesting and reasonable approach for selecting outcomes for PASC, a condition that still lacks a clear definition and mechanistic subtyping.

In the methods section for latent class analysis, there should be a description of how the assumptions of indicator variable independence was tested, and if needed, addressed.

We thank the reviewer for raising an important point. When using traditional latent class analysis (LCA), for example in the setting of item response theory (IRT) or to group survey participants across multiple items, an important step is to evaluate the assumption of local independence i.e. to test that individual items are independent conditional on latent class.

In this study, we used latent class mixed models (LCMMs), a different application also called group-based trajectory modeling. There are no indicator variables to evaluate local independence. Instead, we use linear mixed models to model longitudinal trajectories of ordinal outcomes, and these trajectories form the basis for the latent classes. In this kind of application there is no analogous test of independence, and thus we did not perform the tests as suggested by the reviewer. We have included relevant references in the main text to detailed presentations of the LCMM methodology.

The authors note that “In addition, we did not attempt to identify alternative causes of persistent or new symptoms. However, PRO measures were chosen to attempt to mitigate this limitation by including comparison to pre-illness baseline or some other appropriate recall period when possible.” In a cohort, such as the one described, you would expect new onset symptoms over a year, even without COVID-19. It would be valuable to 1) follow up with review or medical records or survey questions to inquire about alternative diagnosis that might explain PROs and/or at least 2) note incidence of new onset or worsening PROs from literature on post-hospitalization.

This is a valid point and we acknowledge it under limitations. However, we state that PRO measures were chosen to attempt to mitigate this limitation by including comparison to pre-illness baseline or some other appropriate recall period when possible. We did not review medical records to inquire about alternative diagnoses. We did as requested by the reviewer provide 2 studies with data on incidence of new onset or worsening health related quality measures. The data is in line with our findings and we added this information in the text under the limitations section.

Supplement page 3 link for code goes to a page that says “Repository is not available”

We have corrected the link to available code:

https://bitbucket.org/kleinstein/impacc-public-code/src/master/convalescent_manuscript/

Minor comments

In Methods - Statistics, and in the legend for Table 1, the authors state that t-statistics were “recoded such that positive values indicate a greater degree of patient-reported deficit”. However, the minimal deficit (MIN) group has the highest positive values. Should this say “recoded such that negative values...”?

Negative values indicate greater degree of deficit. The text is now corrected.

Under Methods, Data Collection, Study Variables, and Biologic Samples, the complications do not appear to be defined in the previous IMPACC paper or the IMPACC protocol. In particular, given the results highlighted, it would be valuable to know whether patients had secondary bacterial infections, sepsis and/or kidney injury.

We have added the number of participants with these specific complications and any of the other most common complications reported in our previous paper.

The study would be strengthened with information about whether patients had been on medications that affect the frequency of circulating B lymphocytes (rituximab, or other anti-CD20 agents).

There were a total of n=5 participants receiving anti-CD20 medications among the n=590 participants in this convalescent cohort i.e. less than 1%. These participants were distributed across several convalescent clusters (n=2 MIN, n=1 PHY, n=2 COG), all of which received rituximab. No participants received any other anti-CD20 medications. We judged these frequencies to be too low to incorporate this variable into our analyses.

The study would be strengthened by comparison to controls: 1) controls who did not have COVID, 2) patients hospitalized for a non-COVID-19 respiratory viral infection and/or 3) patients hospitalized for elective procedures where the length of stay is similar to PASC. While prospective longitudinal studies may be outside the scope of the current study, it would be helpful to note the limitations of not having these controls when drawing conclusions. For example, conclusions about methylhistidine pathways and long-term patient centered outcomes might reflect something specific to PASC, or factors common to many hospitalizations (stress metabolism, change in muscle tissue, decreased renal function).

We thank the reviewer for the valuable comment and have noted this among our limitations.

For supplemental figures 4S-9S it would be helpful to use asterisks * or some similar convention to highlight which results are significant.

*We thank the reviewer for this suggestion and we replaced P-values with asterisks * to highlight significant results.*

As a side note, degenerate clusters might be particularly interesting as outlier cases for study.

This is an excellent suggestion for a future study, although outside the scope of our current investigation.

Reviewer #3

- Thank you for the opportunity to review this paper! The authors describe a focused study in which they created endotypes of long-Covid and assessed whether certain demographics, clinical characteristics, and viral assays were associated with cluster membership. A number of robust statistical methods were employed to handle the high-dimensional data. The methods & results are well-described. The paper is easy to read. I find no significant methodological limitations or flaws. I applaud the authors on focusing so much on patient-reported outcomes - there is a significant need for more research in this area.

We thank the reviewer for this helpful encouragement.

I have some minor comments/questions below:

- Study Participants - can you add the dates of the study to provide more context of when in the pandemic the study occurred?

Yes of course. This was added to this section.

- The methods seem reasonable, and it's clear a lot of thought went into the best way to handle the complicated dimensionality. Did you consider using a group-based trajectory model for grouping the longitudinal data, and if so, can you speak to why you didn't use that approach instead?

We used group-based trajectory models in the first step of analysis, described in our text as 'latent class mixed models (LCMMs)'. We have added to the text to clarify that our approach is indeed a group-based trajectory model.

- Results - I think you're missing a number regarding the age & BMI where you have written "57 years (IQR 19)," & "31.8 kg/m² (IQR 9.6)", respectively - what's the range? It looks like this was done for length of stay, too - please list the 25th & 75th quantiles when reporting the IQR.

As suggested, IQR values in the main text and Table 1 are now reported with 25th and 75th quantiles.

- Can you provide examples of what counts as an inpatient "complication"?

As noted above, we have now added frequencies of the complications most commonly reported among the inpatient cohort. Examples include acute renal injury/failure, anemia, or acute venous thromboembolism (VTE).

- In your statistical analysis plan, you noted using a multinomial logistic regression for the demographic clinical characteristics associated with each group; however, in Figure 2, you called this a proportional odds model. Can you clarify which regression method you used?

We used a multinomial logistic regression model. The caption for Figure 2 is now corrected.

- In Figure 3, you have a black horizontal line with a number between several clusters - can you state what this number/line represents?

The black horizontal lines denote specific pairwise comparisons between clusters. Following a suggestion above, we now use asterisks instead of numeric p-values to indicate statistical significance.

** See Nature Portfolio's author and referees' website at www.nature.com/authors for information about policies, services and author benefits.

This email has been sent through the Springer Nature Tracking System NY-610A-NPG&MTS

Confidentiality Statement:

This e-mail is confidential and subject to copyright. Any unauthorised use or disclosure of its contents is prohibited. If you have received this email in error please notify our Manuscript Tracking System Helpdesk team at <http://platformsupport.nature.com> .

Details of the confidentiality and pre-publicity policy may be found here <http://www.nature.com/authors/policies/confidentiality.html>

Privacy Policy | Update Profile

Reviewers' Comments:

Reviewer #1:

Remarks to the Author:

I thank the authors for addressing my comments. I am happy with the explanations and revisions.

Reviewer #2:

Remarks to the Author:

The authors have strengthened the paper with additional details on methods, limitations and results. They have also provided a working link to their code repository.

To clarify earlier feedback, it would be interesting to note incidence of new onset or worsening PROs from a control cohort that included post-acute sequelae in patient who had been hospitalized _without_ COVID-19. However, it is understandable that may be outside the scope of this manuscript.

We encourage consideration of including the note on anti-CD20 medications in the supplementary appendix, for those researchers who may be interested.

Reviewer #3:

Remarks to the Author:

The author team did an excellent job addressing my concerns. My only remaining concern is the clarification of the LCMM vs GBTM - the additional phrase placed in the Methods section for the statistical analysis adds to the confusion, I believe. My question focused more on what assumptions did you make about the data (and/or population) as well as the focus (on distinct groups vs. relationships between groups) that lead you to choose the specific modeling approach you used. To clean this up, you might simply consider being more explicit in your modeling choice rationale (e.g., "Because we were focused on ..., we chose to use a [model] that assumes ...")

Reviewer #4:

Remarks to the Author:

The manuscript submitted by Ozonoff and the IMPACC network constitutes an important contribution describing the characterization of post-acute disease followup in a virus naive population, which could be also the future scenario for upcoming pandemics. The manuscript sounds methodologically well described and enough supplementary information is provided by the authors. As mentioned by the authors, stronger and more reproducible results could be reported when relevant data such as number of re-infections, new variants of interest, vaccination status, as well as a longer followup are included. PASC is a complex entity and what factors are influencing it the most, is still under investigation. There is a consistent lack of socioeconomic information from people affected with PASC, which could be driving some outcomes in certain demographic populations. In my opinion, the main limitation is the lack of negative controls (easy to find and recruit at the time of the conception of this study), as well as the lack of blood samples collected at 12 months post-recovery. For definitely attribute PASC disease burden to COVID-19, or to pre-existing conditions worsened by the virus. While briefly discussed the implication of methylhistidine in PASC, the role of acylcarnitines in PHV cluster was not discussed. Further discussion could be provided about the predominant association of SARS-CoV-2 initial burden with certain clusters and the lack of association with the rest of the clusters. The authors must discuss what a functional antibody antiviral activity is considered and its association with baseline characteristics.

Reviewer #5:

Remarks to the Author:

This is an original methodologically sound, unique (by nature of the patients sampled and assays performed), succinctly and accurately written report on a multiparameter observational study of patients self reporting post acute sequelae of the SARS-COV-2 phenotypes and blood sampled at 5 -6 timepoints at and after hospitalization for acute SAVRS-CoV-2 infection. No baseline samples are understandably obtained from these patients in this multi center study that takes in a diverse array of hospitals and institutions. The readout here points to associations of higher respiratory SARS-CoV-2 viral burden and lower Receptor Binding Domain and Spike antibody titers, and multidomain deficit clusters including a lower frequency of circulating B lymphocytes by mass cytometry (CyTOF) elevated circulating fibroblast growth factor 21 (FGF21) especially in the mental / cognitive area. These are statistically significant and noteworthy from this observational study that are delivered with all the caveats and limitations that the authors openly detail.

This work is incrementally informative to the important growing field of the pathogenesis of long COVID and clearly opens the way to all important mechanistic studies to understand diagnosis, treatment options and prognosis. The limitation here is significant here and relates to the sample collection, the self reporting symptom approach, the absence of baseline data and then ultimately what this combination of findings (including anti spike antibody, B cell, FGF21) actually mean in real terms without a validation cohort and some handle on mechanism here.

Thank you for allowing us the opportunity to revise the resubmission of our paper entitled "Features of acute COVID-19 associated with Post-Acute Sequelae of SARS-CoV-2 phenotypes: results from the IMPACC study". Below in *red italics* is our point-by-point responses to editorial and reviewer comments.

Reviewer #1 (Remarks to the Author):

I thank the authors for addressing my comments. I am happy with the explanations and revisions.

We thank the reviewer for their time.

Reviewer #2 (Remarks to the Author):

The authors have strengthened the paper with additional details on methods, limitations and results. They have also provided a working link to their code repository.

To clarify earlier feedback, it would be interesting to note incidence of new onset or worsening PROs from a control cohort that included post-acute sequelae in patient who had been hospitalized *_without_* COVID-19. However, it is understandable that may be outside the scope of this manuscript.

We encourage consideration of including the note on anti-CD20 medications in the supplementary appendix, for those researchers who may be interested.

We thank the reviewer for their comment. Unfortunately, we do not have access to PRO data on patients hospitalized without COVID-19.

We have included the following under supplementary results: "Of note, there were a total of 5 participants receiving anti-CD20 medications among the n=590 participants in this convalescent cohort i.e. less than 1%. These participants were distributed across several convalescent clusters (n=2 MIN, n=1 PHY, n=2 COG), and all 5 received rituximab. No participants received any other anti-CD20 medications".

Reviewer #3 (Remarks to the Author):

The author team did an excellent job addressing my concerns. My only remaining concern is the clarification of the LCMM vs GBTM - the additional phrase placed in the Methods section for the statistical analysis adds to the confusion, I believe. My question focused more on what assumptions did you make about the data (and/or population) as well as the focus (on distinct groups vs. relationships between groups) that lead you to choose the specific modeling approach you used. To clean this up, you might simply consider being more explicit in your modeling choice rationale (e.g., "Because we were focused on ..., we chose to use a [model] that assumes ...")

We thank the reviewer for their comment and have made the recommended edit.

Reviewer #4 (Remarks to the Author):

The manuscript submitted by Ozonoff and the IMPACC network constitutes an important contribution describing the characterization of post-acute disease followup in a virus naive population, which could be also the future scenario for upcoming pandemics. The manuscript sounds methodologically well described and enough supplementary information is provided by the authors. As mentioned by the authors, stronger and more reproducible results could be reported when relevant data such as number of re-infections, new variants of interest, vaccination status, as well as a longer followup are included. PASC is a complex entity and what factors are influencing it the most, is still under investigation. There is a consistent lack of socioeconomic information from people affected with PASC, which could be driving some outcomes in certain demographic populations. In my opinion, the main limitation is the lack of negative controls (easy to find and recruit at the time of the conception of this study), as well as the lack of blood samples collected at 12 months post-recovery. For definitely attribute PASC disease burden to COVID-19, or to pre-existing conditions worsened by the virus. While briefly discussed the implication of methylhistidine in PASC, the role of acylcarnitines in PHV cluster was not discussed. Further discussion could be provided about the predominant association of SARS-CoV-2 initial burden with certain clusters and the lack of association with the rest of the clusters. The authors must discuss what a functional antibody antiviral activity is considered and

its association with baseline characteristics.

We thank the reviewer for their comments. Because of restrictions on research activities and facilities when this cohort was initiated in the spring of 2020, it was not possible to enroll a concurrent cohort with mild COVID-19, or healthy controls. We do not have neutralizing antibody activity on the whole cohort and therefore cannot make an association with baseline characteristics. We believe that neutralizing antibodies and binding antibodies have strong correlation in this unvaccinated cohort enrolled when very few variants of interest or concern circulated based on prior publication [Rapid Generation of Neutralizing Antibody Responses in COVID-19 Patients - PMC (nih.gov)]. While neutralizing activity is a correlate of protection (e.g. <https://pubmed.ncbi.nlm.nih.gov/34210573/> and <https://www.science.org/doi/10.1126/science.abm3425>) it has been shown that binding antibodies also correlate with protection (e.g. <https://pubmed.ncbi.nlm.nih.gov/34210573/>) even in the absence of strong neutralizing activity (e.g. <https://www.sciencedirect.com/science/article/pii/S2666776223000650>) potentially through Fc-FcR interactions. We also added discussion about acylcarnitines and its possible implications to long covid outcomes as suggested by the reviewer.

Reviewer #5 (Remarks to the Author):

This is an original methodologically sound, unique (by nature of the patients sampled and assays performed), succinctly and accurately written report on a multiparameter observational study of patients self reporting post acute sequelae of the SARS-COV-2 phenotypes and blood sampled at 5 -6 timepoints at and after hospitalization for acute SAVRS-CoV-2 infection. No baseline samples are understandably obtained from these patients in this multi center study that takes in a diverse array of hospitals and institutions. The readout here points to associations of higher respiratory SARS-CoV-2 viral burden and lower Receptor Binding Domain and Spike antibody titers, and multidomain deficit clusters including a lower frequency of circulating B lymphocytes by mass cytometry (CyTOF) elevated circulating fibroblast growth factor 21 (FGF21) especially in the mental / cognitive area. These are statistically significant and noteworthy from this observational study that are delivered with all the caveats and limitations that the authors openly detail.

This work is incrementally informative to the important growing field of the pathogenesis of long COVID and clearly opens the way to all important mechanistic studies to understand diagnosis, treatment options and prognosis. The limitation here is significant here and relates to the sample collection, the self reporting symptom approach, the absence of baseline data and then ultimately what this combination of findings (including anti spike antibody, B cell, FGF21) actually mean in real terms without a validation cohort and some handle on mechanism here.

We thank the reviewer for their comprehensive review, we agree and acknowledge the limitations of our work.